**Technology**

# ChromBERT: A foundation model for learning interpretable representations for context-specific transcriptional regulatory networks

## Graphical abstract

## Authors

Zhaowei Yu, Dongxu Yang,
Qianqian Chen, ..., Yucheng Wang,
Chenfei Wang, Yong Zhang

## Correspondence

yzhang@tongji.edu.cn

## In brief

Yu et al. present ChromBERT, a genomic foundation model pre-trained on ~1,000 transcription regulators to learn their general interaction syntax. ChromBERT accurately imputes regulatory landscapes and interprets context-specific regulatory networks in unseen cell types and single cells, identifying key drivers of cell state transitions without additional ChIP-seq experiments.

## Highlights

- A pre-trained foundation model learns interaction syntax of ~1,000 transcription regulators

- ChromBERT imputes regulatory landscapes in unseen cell types and single cells

- Fine-tuning captures context-specific regulatory effects and dynamics

- ChromBERT interprets key regulators driving cell state transitions without ChIP-seq experiments

 Yu et al., 2026, Cell Genomics 6, 101130
April 8, 2026 © 2025 The Author(s). Published by Elsevier Inc.

# Cell Genomics

CellPress

## Technology

# ChromBERT: A foundation model for learning interpretable representations for context-specific transcriptional regulatory networks

Zhaowei Yu,[1,2] Dongxu Yang,[1,2] Qianqian Chen,[1,2] Yuxuan Zhang,[1,2] Zhanhao Li,[1] Yucheng Wang,[1] Chenfei Wang,[1] and Yong Zhang[1,3,*]

[1]State Key Laboratory of Cardiovascular Diseases and Medical Innovation Center, Institute for Regenerative Medicine, Department of Neurosurgery, Shanghai East Hospital, Shanghai Key Laboratory of Signaling and Disease Research, Frontier Science Center for Stem Cell Research, School of Life Sciences and Technology, Tongji University, Shanghai 200092, China
[2]These authors contributed equally
[3]Lead contact
*Correspondence: yzhang@tongji.edu.cn

## SUMMARY

Gene expression is shaped by transcriptional regulatory networks (TRNs), where transcription regulators interact within regulatory elements in a context-specific manner. Deciphering context-specific TRNs has long been constrained by the severe sparsity of cell-type-specific chromatin immunoprecipitation sequencing (ChIP-seq) profiles. Here, we present ChromBERT, a foundation model pre-trained on large-scale human ChIP-seq datasets covering ~1,000 transcription regulators. ChromBERT learns the genome-wide syntax of regulatory cooperation and generates interpretable TRN representations. After prompt-enhanced fine-tuning, it outperforms existing methods for imputing unseen cistromes. Moreover, lightweight fine-tuning on cell-type-specific downstream tasks adapts the TRN representations to capture regulatory effects and dynamics within any given cellular context. The resulting context-specific representations can then be interpreted to infer regulatory roles of transcription regulators underlying these cell-type-specific regulatory outcomes without requiring additional ChIP-seq experiments. By overcoming the limitations of sparse transcription regulator data, ChromBERT significantly enhances our ability to model and interpret transcriptional regulation across a wide range of biological contexts.

## INTRODUCTION

Gene expression patterns are shaped by the intricate interactions among transcription regulators, regulatory elements, and target genes, which together form transcriptional regulatory networks (TRNs).[1–4] A central goal in biology is to understand how transcription regulators interact within individual genomic regions, how distinct regulatory architectures are formed across the genome, and how these interactions vary across diverse cell types. Landmark projects like the Encyclopedia of DNA Elements (ENCODE) have provided invaluable insights into transcriptional regulatory landscapes by generating extensive data on transcription factor binding and chromatin states.[2,4] However, these efforts have primarily focused on a limited number of well-studied cell types. Consequently, the context-dependent interactions of transcription regulators and their hierarchical roles in modulating gene expression remain poorly understood in most biological settings.

Pre-trained foundation models have achieved remarkable success in genomics studies by leveraging large-scale heterogeneous datasets to learn fundamental principles and then transferring this knowledge to a wide range of downstream tasks.[5–13] The representations learned during pre-training can be interpreted using post hoc analyses or by-design strategies to uncover valuable biological insights.[14] Task-specific fine-tuning expands this capability by adjusting the learned representations for specific tasks, generating biological insights that are directly relevant to the given fine-tuning objective.[6,10,15] By combining broad generalization from pre-training with task-specific adaptability through fine-tuning, pre-trained foundation models are particularly suitable for representing and interpreting the complex dynamics of TRNs, especially in settings where transcription regulator landscapes are highly incomplete and poorly characterized.

Foundation models pre-trained on DNA sequences have demonstrated remarkable capabilities to decipher the regulatory architecture of genomes. They effectively capture the organizational syntax of genomic sequences and decode the *cis*-regulatory grammar that governs gene transcription.[7,8,11,12,16] Recent models, such as EpiBERT[17] and EPCOT,[18] integrate DNA sequence features with chromatin accessibility profiles to generalize sequence-inferred regulatory grammar across different cell types. However, these models often fall short in addressing the roles of *trans*-acting transcription regulators, which are critical

for a comprehensive understanding of gene transcription. A recent study seeks to bridge this gap by integrating DNA-binding motifs of transcription regulators and chromatin accessibility to improve gene expression predictions,[10] and another study also integrates motifs with multi-omics data to infer gene regulatory architecture.[19] However, these motif-centric approaches are limited as accessible motifs account for only a fraction of actual binding events, and many transcription regulators lack well-characterized motifs.[20,21] Binding events are further modulated by co-factors, chromatin context, and epigenetic modifications, which extend beyond simple motif recognition. Consequently, motif-dependent approaches are insufficient to capture the intricate and context-specific interactions among transcription regulators, highlighting the need for foundation models that learn these complex interactions directly.

Here, we introduce ChromBERT, a foundation model specifically designed to directly model genome-wide combinatorial binding patterns of transcription regulators. Pre-trained on the Cistrome-Human-6K dataset, comprising large-scale, cell-type-agnostic ChIP-seq data, ChromBERT effectively learns the interaction syntax of transcription regulators across the genome (Figure 1A). By leveraging advanced transformer architectures, ChromBERT generates interpretable representations of TRNs and of individual regulators at each genomic region, enabling the interpretation of genome-wide regulatory roles and functional collaborations of these regulators. ChromBERT demonstrates superior performance in cistrome imputation for unseen cell types through prompt-enhanced fine-tuning, outperforming previous methods, and showcasing robust generalization across a diverse range of cell types. When fine-tuned on cell-type-specific downstream tasks, the model not only achieves superior predictive performance but also adapts its representations to reflect the unique regulatory architecture of each specific cell type. This task-specific fine-tuning provides clear, context-specific interpretability into the roles of transcription regulators in driving observed regulatory effects and dynamic changes within the given cellular context, without requiring additional cell-type-specific ChIP-seq data for each regulator (Figure 1B). This capability is particularly significant in addressing the challenges posed by sparse transcription regulator data, a common limitation in most biological settings where comprehensive datasets for transcription regulators are often unavailable.

## DESIGN

ChromBERT distinguishes itself from other genomics foundation models that primarily focus on DNA sequence organization[7,8] by treating genome-wide ChIP-seq tracks of transcription regulators as a "language" of regulatory interactions. Each genomic region is encoded as an ordered vector of binding statuses for all transcription regulators, enabling the model to learn their intricate interaction syntax within each genomic region (Figure 1A). This contrasts with current genomics foundation models, which predominantly model the syntax of the neighboring DNA sequence elements or interactions between adjacent genomic regions. We pre-trained ChromBERT on a large-scale corpus of transcription regulator landscapes using a masked learning

strategy akin to BERT[22] and self-attention mechanisms.[23] This self-supervised training gives ChromBERT an interpretable, genome-wide understanding of TRNs that can be transferred effectively to a variety of downstream transcription regulation tasks.

The pre-training dataset for ChromBERT was constructed by compiling and filtering nearly all publicly available human ChIP-seq, DNase-seq, and assay for transposase-accessible chromatin using sequencing (ATAC-seq) data from the Cistrome Data Browser[24] (see STAR Methods). This process resulted in the Cistrome-Human-6K dataset, comprising 6,391 qualified cistromes (genome-wide maps of the *cis*-regulatory binding sites of *trans*-acting regulators[25]) for 991 transcription regulators, 76 histone modifications, and chromatin accessibility from various cell types (Figure 1A; Table S1). Binding signals from all cistromes were discretized into binding statuses, which were then aggregated over each 1-kilobase (kb) genomic interval to construct input features for ChromBERT (Figures S1A and S1B). During pre-training, 15% of these binding statuses were randomly masked and had to be reconstructed from the remaining cistromes of the same 1-kb genomic interval. This approach enabled the model to learn interaction syntax among transcription regulators across the genome. Given the limited number of cistromes in individual cell types (Figure S1C), we pooled all qualified human cistromes across cell types to enrich the diversity of input features. While this cell-type-agnostic strategy introduces variability in representation frequencies for regulators and cell types, it maximizes the use of all available data and supports the model's generalization ability. Pre-training on over two million such 1-kb genomic regions enhanced ChromBERT's capacity to capture intricate interaction patterns and achieve strong awareness of diverse genomic contexts. For each genomic region presented to ChromBERT, the pre-trained model embeds it into a TRN embedding that represents the combinatorial interactions among over one thousand transcription regulators within the given region (Figure 1A; see STAR Methods). Each regulator's embedding, encoded as a 768-dimensional vector and aggregated from its individual cistromes, represents its contextual interactions with other regulators in the given region. Subsequent fine-tuning on cell-type-specific tasks adapts these embeddings to the target cellular context, providing clear, context-specific insights into the regulators that drive the observed regulatory outcomes without requiring additional cell-type-specific genomic data (Figure 1B).

## RESULTS

### Pre-trained embeddings reveal human regulatory architecture

After pre-training, we interpreted the pre-trained embeddings from ChromBERT to check whether they could effectively reveal the transcriptional regulatory architectures specific to their genomic regions. Given the diversity of transcriptional regulatory architectures across the genome, we hypothesized that pre-trained TRN embeddings would cluster similarly according to known groups of regulatory elements. Applying ChromBERT to groups of regulatory elements on chromosome one, as defined by ChromHMM states,[26] we observed that TRN embeddings

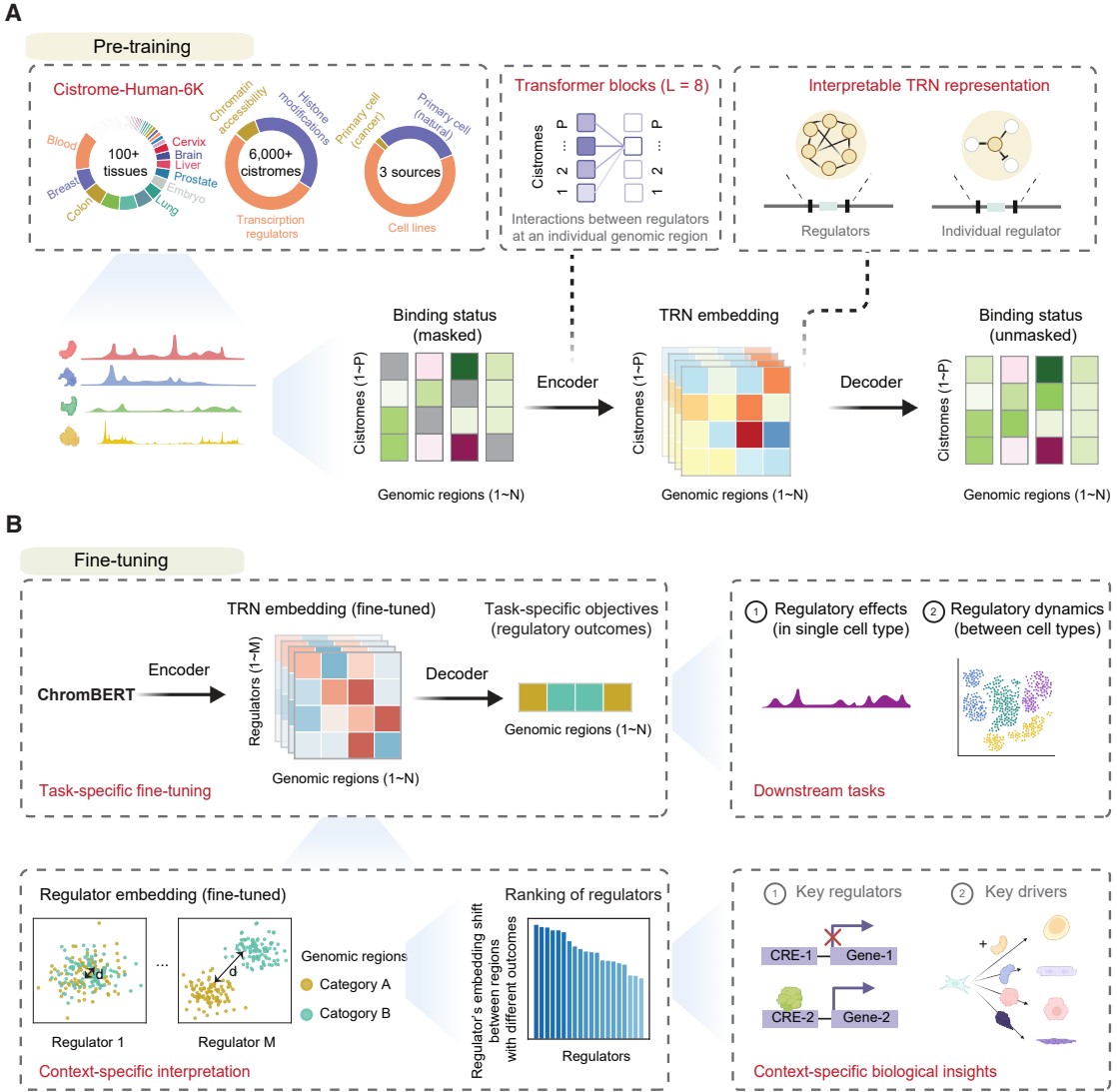

**Figure 1. The ChromBERT architecture**

(A) Pre-training schematic of ChromBERT. ChromBERT is first pre-trained on the Cistrome-Human-6K dataset, which encompasses cistromes derived from over 100 tissues, sourced from three primary categories: cell lines, primary cancer cells, and naturally occurring primary cells. The input is a binding status matrix, representing the binding statuses of cistromes (*y* axis in the heatmap, 6,391 input features) across genome-wide 1 kb intervals (*x* axis in the heatmap, 2,137,894 training samples). Fifteen percent of matrix entries are randomly masked (gray in the heatmap), and an eight-layer transformer encoder converts this partially masked matrix into TRN embeddings. A lightweight decoder then reconstructs the masked values, forcing the model to learn the interaction syntax among regulators (see STAR Methods). The TRN embedding represents the transcriptional regulatory network (the functional interplay between regulators) at each individual genomic region and can be further decomposed into per-regulator embeddings that capture the role of each factor within that local network. See also Figures S1–S3.

(B) Fine-tuning schematic of ChromBERT. *Task-specific fine-tuning*: ChromBERT can be fine-tuned for various transcription regulation-related downstream tasks, such as predicting regulatory effects in a single cell type or regulatory dynamics between pairs of cell types. In each case, the model is trained on genome-wide regulatory outcomes that either classify genomic regions into discrete categories (classification) or assign them continuous values (regression). *Context-specific interpretation*: fine-tuning adapts ChromBERT's TRN embeddings to the given cellular contexts, allowing for the interpretation of transcription regulators' context-specific roles. The key principle is that when regulator embeddings shift systematically between regions with different outcomes, the regulators showing the largest shifts are the most likely drivers of the change. Ranking these shifts thus reveals the key transcription regulators that control cell-type-specific regulation or cell state transitions.

generally clustered separately (Figure S2A). Moreover, within a given cell type, accessible promoter-proximal and distal regions exhibited markedly different embedding patterns (Figure S2B). These results suggest that the embedding effectively delineates

distinct regulatory architectures. We also hypothesized that TRN embeddings of adjacent genomic regions would display higher similarity due to shared local chromatin structure, even though ChromBERT does not model interactions between regions.

**Figure 2. ChromBERT boosts cistrome imputation**

(A) Schematic of cistrome imputation in new cell types or single cells by prompt-enhanced fine-tuning of ChromBERT. This process utilizes TRN embedding (representing region-specific regulatory architecture) along with cell-type-specific prompts (representing cell types being imputed) and regulator embedding (representing regulators being imputed) (see STAR Methods).

(B) Performance comparison for imputing 241 cistromes in cell lines and primary cells. The comparison is between ChromBERT-D (ChromBERT with DNase-seq prompts), and a baseline method that utilize DNase-seq signals and regulator-specific candidate peaks (see STAR Methods). The top heatmap shows performances using the AUPRC metric for both approaches, while the bottom bar plot illustrates performance differences. Red bars indicate ChromBERT-D's superiority, while blue bars indicate the baseline method's advantage. All test cistromes were excluded from the training dataset for prompt-enhanced fine-tuning.

(C) Heatmaps showing predictive probability of ChromBERT-D (red), alongside ChIP-seq signals (green) and DNase-seq signals (blue) on cell-type-specific peaks of BRD4 that are accessible in both K562 and A549 (see STAR Methods). The BRD4 ChIP-seq and DNase-seq signals were normalized to the genome background. The black dashed lines divide the test regions into K562 highly-specific, weakly-specific, and A549 highly-specific peaks, arranged from bottom to top.

(D) The histogram illustrates performance of ChromBERT-D on cell-type-specific peaks with similar chromatin accessibility (both accessible regions). The metrics were calculated by using the difference in predictive probabilities between two cell types to classify the ground truth of cell type specificity obtained from ChIP-seq peaks. This evaluation included 186 cell-type pairs for 24 regulators.

(E) The UCSC genome browser view illustrates the representative BRD4 binding sites (green) in A549 and K562 cell lines, at consistent peaks (hg38:chr17:28,351,650-28,365,238), A549-specific peaks (hg38:chr1:220,683,084-220,697,918), and K562-specific peaks (hg38:chr3:14,334,778-14,370,222)

*(legend continued on next page)*

As expected, embedding similarity decreased with increasing distance between regions (Figure S2C). Overall similarity in this analysis remained relatively high, likely because a large fraction of the genome comprises inactive regions with limited regulator binding. Consequently, many regions share broadly similar regulatory contexts with low discriminative power, leading to higher averaged cosine similarity than would be expected from random embeddings.

Considering the impact of 3D chromatin organization, which segregates the genome into potential functional units,[27] we examined the association between TRN embeddings and 3D genome architecture. Our results showed that regions within the same topologically associating domain (TAD) exhibited higher similarity in their embeddings compared to those separated by TAD boundaries (Figure S2D). Furthermore, long-range genomic region pairs that exhibited chromatin contacts showed significantly higher embedding similarity than those without contacts (Figures S2E–S2G). To further evaluate how TRN embeddings relate to 3D genome organization, we trained a deep learning model that integrates embedding similarity with Hi-C contact data from long-range genomic region pairs to impute higher-resolution chromatin contacts (see STAR Methods). This approach successfully imputed Micro-C dataset in human embryonic stem cells and demonstrated robust generalization to unseen cell types (Figure S2H). These findings demonstrate that ChromBERT's pre-trained TRN embeddings effectively represent the complex regulatory architecture of the human genome and distinguish different genomics regions.

Next, we assessed how well ChromBERT's regulator embeddings capture functional cooperation among transcription regulators. Each regulator embedding at a given genomic region encapsulates its contextual relationship to all other regulators. Intuitively, we reasoned that regulators that functionally collaborate should have similar embeddings. To test this, we averaged each regulator's embeddings across genomic regions on chromosome one and calculated their pairwise similarities. This analysis revealed a highly heterogeneous pattern of co-association among transcription regulators, forming distinct clusters (Figure S3A). We identified one cluster involving well-characterized transcription factors and co-activators, such as SMARCA4, EP300, BRD4, MED1, FOXM1, and MYC (Figure S3A), highlighting that regulators with similar embeddings tend to participate in the same functional network. Regulators' embedding similarities also correlated with more systematic measures of functional cooperation. Regulator pairs with high embedding similarity showed elevated protein-protein interaction frequencies (as defined by affinity-purification mass spectrometry data from BioPlex 3.0[28]) and functional association (as defined by hallmark gene sets and ontology gene sets from the Molecular Signatures Database[29]) (Figure S3B). We next examined embedding similarity between regulators across genomic contexts to evaluate whether the model captures context-specific regulatory relationships. RNF2, a core component of polycomb repressive complex 1 (PRC1),[30] shows high embedding similarity to classical repressors such as PCGF2 in repressive chromatin regions, but shifts toward transcriptional activators (e.g., EP300 and MYC) in regions associated with active regulation (Figure S3C). This context-dependent switching mirrors RNF2's documented dual roles and demonstrates that the learned regulator embeddings encode biologically meaningful, context-specific regulatory associations. Randomly shuffling input co-binding patterns abolished these associations, confirming that the observed embedding similarities arise from true ChIP-seq binding relationships rather than global co-occurrence biases (Figure S3D). Together, these results show that ChromBERT's embeddings faithfully encode context-dependent functional cooperation among transcription regulators.

## ChromBERT boosts cistrome imputation in unseen cell types

Despite extensive experimental profiling, the genomic landscapes of most transcription regulators remain far from comprehensive. We therefore fine-tuned ChromBERT to impute cistromes of transcription regulators, i.e., to predict the presence or absence of binding events, in cell types that lack ChIP-seq data. A key challenge is to incorporate cell type information so that the model can transfer across diverse cell types. Inspired by the advances in prompt-enhanced fine-tuning, a highly efficient and flexible approach that has proven transformative in pre-trained large language models,[31–34] we designed cell-type-specific prompts (from chromatin accessibility profiles or transcriptome profiles, respectively) to guide the model (see STAR Methods). Previous studies have shown that both chromatin accessibility and transcriptome profiles are strongly linked to transcription regulator binding specificity.[35,36] These prompts integrate with ChromBERT's regulator embeddings through concatenation, enabling flexible and accurate imputation for distinct regulators across diverse cell types (Figure 2A). To align prompts and embeddings within a unified feature space, we fine-tuned the model using approximately one hundred cistromes from multiple cell types, supplemented by paired DNase-seq data or scGPT[6] cell prompts generated from RNA-seq data in the related cell types (see STAR Methods). After this alignment, ChromBERT can impute any cell type-regulator pair without further training.

ChromBERT-D (DNase-seq prompts) demonstrated superior performance in cistrome imputation across a wide range of bulk cell types by leveraging DNase-seq data as prompts. It achieved a mean area under the precision-recall curve (AUPRC) of 0.554 on test cistromes, significantly outperforming the baseline approach (AUPRC = 0.271), which integrates regulator-specific candidate peaks and DNase-seq profiles and is

---

arranged from left to right. Predictive probabilities of ChromBERT-D for BRD4 (red) are shown separately for each cell type, alongside the corresponding DNase-seq signals (blue).

(F) Scatter plots show the performance comparison between ChromBERT-D and Avocado (see STAR Methods) across 57 cistromes. The diagonal gray line indicates equal performance, with percentage of cistromes where ChromBERT-D outperforms and the statistical significance calculated by a two-sided Student's *t* test were annotated.

See also Figures S4 and S5.

a common strategy in the absence of ChIP-seq data. This performance advantage held for both cell lines and primary cells (Figure 2B). ChromBERT-D also exhibited exceptional robustness, with its performance being largely insensitive to the varying representation frequency of individual cell types or regulators in the pre-training data, except for a small subset of regulators that were massively over-represented (Figures S4A and S4B). Notably, one of the most challenging aspects of cistrome imputation is identifying cell-type-specific binding sites, which are biologically significant yet computationally challenging, especially when chromatin accessibility is similar across cell types. Importantly, ChromBERT-D demonstrated superior performance in classifying cell-type-specific binding sites that could not be explained by differential chromatin accessibility alone (Figures 2C–2E and S4C). Furthermore, when benchmarked against Avocado, a state-of-the-art model for cistrome imputation,[37] ChromBERT-D achieved an AUPRC of 0.659 versus 0.514 for Avocado across applicable test sets (Figure 2F). ChromBERT-R (RNA-seq prompts) also performed well using bulk RNA-seq data as prompts (Figure S4D). Although its overall accuracy was lower than ChromBERT-D, it provided complementary predictive power for regulators such as EZH2 and REST, whose activities are not strictly determined by chromatin accessibility (Figure S4E). However, combining both modalities through simple feature concatenation did not improve performance beyond DNase-seq alone, suggesting that more advanced multimodal integration strategies are needed to fully capture the synergy between chromatin accessibility and transcriptional states.

Imputing single-cell cistromes is significantly more challenging because paired single-cell multi-omics data remain scarce.[38,39] Despite the rapid expansion of single-cell transcriptomics over the past decade, the corresponding binding data for transcription regulators are still missing, limiting our view of transcription regulation at the single-cell level. Building on the successful alignment of bulk-cell RNA-seq prompts from scGPT with pre-trained ChromBERT embeddings for bulk-cell cistromes imputation (Figure S4D), we tested its potential to impute cistromes at single-cell resolution by replacing bulk-cell RNA-seq prompts with single-cell prompts (see STAR Methods). Performance was evaluated in human peripheral blood mononuclear cells (PBMCs) single-cell multi-omics dataset using two metrics: predictive accuracy in B cells, using ChIP-seq data from GM12878 as ground truth; and the single-cell-specificity of prediction, using motif accessibility inferred from single-cell ATAC-seq data as ground truth (see STAR Methods). ChromBERT-R achieved significantly higher predictive performance for eight regulators in B cells, outperforming a combination of DNA-binding motifs and imputed chromatin accessibility (Figure S5A). Additionally, ChromBERT-R exhibited robust single-cell-specific performance (Figures S5B and S5C). These results show that ChromBERT-R accurately imputes single-cell cistromes and can be readily adapted to more diverse biological contexts through tailored prompt engineering.

## ChromBERT reveals key regulators in specific cell types

Regulatory effects such as enhancer activity and gene expression levels have already been profiled across many cell types, but the transcription regulators that produce these outcomes remain

largely unknown, because cell-type-specific binding data are scarce. ChromBERT bridges this gap in two steps. First, large-scale pre-training produces embeddings that encode TRNs for every genomic region. Second, task-specific fine-tuning adapts these embeddings by training the model to predict the measured outcome (e.g., enhancer activity or gene expression across the genome) in the given cellular context (Figures 3A and S6). After task-specific fine-tuning, the embeddings acquire context-specific interpretability: for each region, a regulator's embedding summarizes its local interactions with all other regulators. When these embeddings shift systematically between regions with different outcomes, the regulators showing the largest shifts are the most likely drivers of the change (Figure 3A; see STAR Methods). Because this inference depends only on the fine-tuned embeddings, not on additional ChIP-seq experiments, it can reveal key regulators even in rare or experimentally intractable samples. We illustrate the approach by fine-tuning ChromBERT on three downstream tasks and using the resulting context-specific embeddings to pinpoint candidate driver regulators.

The first task involved tissue-specific fine-mapping of expression quantitative trait loci (eQTL). eQTLs are genetic variants that modulate gene expression levels; fine-mapping them is essential for pinpointing the causal variants underlying complex traits and diseases.[45] We fine-tuned ChromBERT to classify causal and non-causal variants using the latest data from the eQTL catalog, incorporating DNA sequence variation via a DNA sequence prompt from DNABERT-2 (see STAR Methods). As shown in Figure 3B, fine-tuned ChromBERT achieved a mean area under receiver operating characteristic (AUROC) of 0.828, outperforming Enformer (0.770), DNABERT-2 model alone (0.788), and ChromBERT without pre-training (0.804) across 49 tissues. After fine-tuning, ChromBERT's contextual embeddings across eQTL loci highlighted regulators and epigenetic modifications that differed significantly between causal and non-causal variants (see STAR Methods). Chromatin accessibility, indicated by DNase-seq signals, showed the largest average shift across 49 tissues (Figure S7A), aligning with its known significant role in transcription regulation.[46,47] Further analysis of public DNase-seq profiles across 10 tissues confirmed that causal eQTLs exhibited elevated chromatin accessibility (Figure S7B). Additionally, we also investigated transcription regulators that exhibited a significant shift in contextualized embeddings between the two groups of eQTLs. These differential regulators not only have a higher motif presence at causal QTLs but also demonstrate significant alterations in motif affinity following single-nucleotide genetic variations at these loci (Figure 3C). For instance, in a case study of causal eQTL rs17079281 influencing the transcription of *DCBLD1* in the lung, we found that a C to T variation led to increased motif affinity of YY1 (Figure 3D), a differential regulator identified by ChromBERT. It is consistent with a previous study demonstrating that rs17079281 decreases lung cancer risk by creating a YY1-binding site to suppress *DCBLD1* expression.[48] These results demonstrate that ChromBERT not only fine-maps eQTLs with superior accuracy, but also provides valuable insights into how specific genetic variants drive gene expression changes.

The second task was to fine-tune ChromBERT for predicting genome-wide enhancer activity quantified from STARR-seq in HCT116 cells.[49] In this task, ChromBERT showed superior

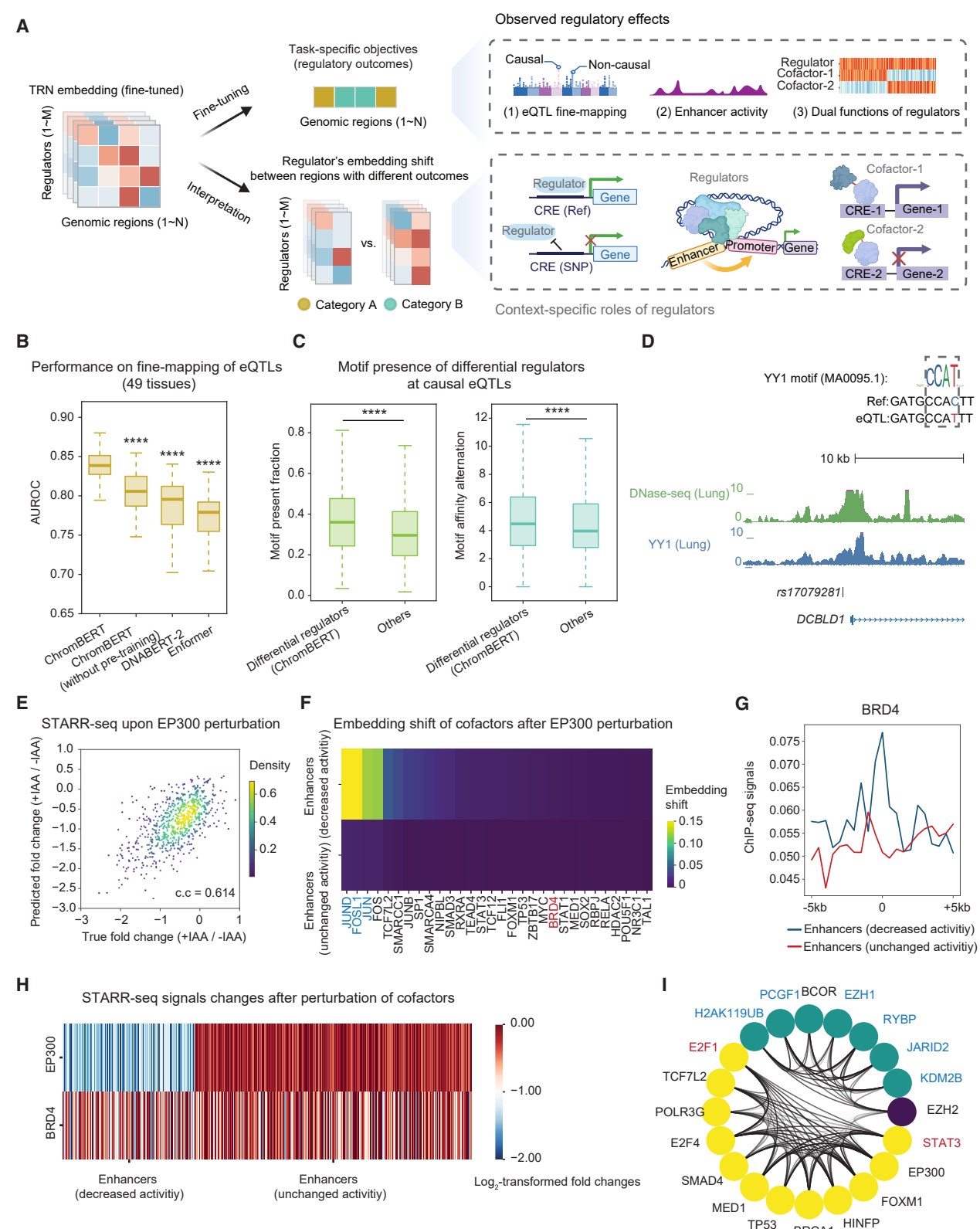

*(legend on next page)*

prediction performance than Enformer, DNABERT-2, and ChromBERT without pre-training (Figure S7C). Although many transcription regulators and cofactors co-occupy active enhancers, perturbation studies show that distinct enhancer classes rely on different cofactor dependencies.[43,50] To explore these dependencies in enhancer activity, we further fine-tuned ChromBERT to predict perturbation effects of regulators using an STARR-seq dataset for EP300/CREBBP, BRD2, and CDK7 perturbations.[43] The model accurately predicted both wild-type and perturbed STARR-seq signals, capturing at least a-fraction of perturbation effects (Figures 3E, S7D, S7E, and S7F). We then used embedding shifts upon perturbations to identify regulators that depend on the perturbed factor. Specifically, at enhancers whose activity dropped after EP300 inhibition, regulators showing significant embedding shifts upon EP300 perturbation were flagged as EP300 cofactors (see STAR Methods). This approach identified cofactors such as FOSL1 and JUND having strong functional associations with EP300 (Figure 3F), in line with the previous study.[43] By contrast, BRD4, one well-characterized cofactor enriched at these enhancers, displayed little embedding shift, suggesting a weaker dependency with EP300 (Figures 3F and 3G). Re-analysis of STARR-seq data after BRD4 perturbation confirmed this weak association (Figure 3H). Together, these results show that fine-tuned ChromBERT not only predicts enhancer activity with high accuracy but also disentangles the complex cofactor dependencies that govern enhancer function.

The third task focused on a multi-functional transcription regulator. EZH2, the catalytic subunit of PRC2,[51] operates in two distinct modes: a classical, H3K27me3-dependent repressive role and a non-classical function, H3K27me3-independent role.[52,53] We fine-tuned ChromBERT to classify EZH2 ChIP-seq peaks in human embryonic stem cells as classical and non-classical, where ChromBERT showed a better performance in this task than all other methods (Figure S7G). Interpreting the task-adapted embeddings revealed two regulator modules that co-associate with EZH2 (Figure 3I). The first module is strongly associated with regulators involved in repressive function, aligning with the classical function of EZH2. In contrast, the second module contains activators, including E2F1 and STAT3, previously implicated in EZH2's non-classical function.[53–55] We next evaluated the robustness of ChromBERT's embedding-based interpretation across all three downstream tasks using alternative similarity and shift metrics beyond cosine similarity and 1-cosine similarity. The results remained largely consistent (Figure S8), indicating that the interpretability framework is insensitive to the specific choice of metric. We also examined the impact of limited training data. While severe down-sampling led to overfitting and reduced validation performance (Figures S9A and S9B), selecting checkpoints with the lowest validation loss mitigated this effect. Under this strategy, ChromBERT maintained stable performance even with moderate reductions in training size (Figures S9C and S9D), demonstrating robustness under data-sparse conditions. These results

**Figure 3. ChromBERT reveals key regulators in cell-type-specific regulatory effects**

(A) Schematic of prompt-enhanced fine-tuning and interpretation for cell-type-specific regulatory effects. From left to upper right: ChromBERT is fine-tuned on a cell type or tissue of interest: a universal decoder learns to predict genome-wide regulatory outcome for each region, either as discrete labels (classification) or continuous values (regression). From left to lower right: the resulting context-specific embeddings are then analyzed. The key principle in interpretation is that regulators whose embeddings shift most markedly between two groups of regions with distinct outcomes are inferred to drive the change (Figure 1B; see STAR Methods). This schematic was created in BioRender. Yu, Z. (2025) https://BioRender.com/3biab7r.

(B) Boxplots comparing the performance in classifying causal versus non-causal eQTLs across 49 human tissues for ChromBERT, ChromBERT (without pre-training), DNABERT-2, and Enformer, as indicated by AUROC (see STAR Methods).

(C) Boxplots comparing (1) fraction with presence of a regulator's motif and (2) motif affinity alternation within ±10 bp of causal eQTLs, between the top 20% of regulators showing the highest embedding shifts and all others. The motif presence ($p$ value < 0.01 in either wild type or mutant form) and affinity (log-odds scores) were determined using FIMO (v5.5.5),[40] and only sites with motif affinity > 0 were analyzed for motif affinity alternation.

For (B) and (C), the statistical significance was performed by a two-sided Mann-Whitney U-test and **** represents $p$ value < $1 \times 10^{-4}$. The center lines mark the median, the box limits indicate the 25th and 75th percentiles, and the whiskers extend to 1.5× the interquartile range from the 25th and 75th percentiles.

(D) The UCSC genome browser view shows YY1 signals (blue) near eQTL rs1707928 (human genome hg38), along with DNase-seq signals in lung (green). The YY1 ChIP-seq signals were from GEO: GSE32465,[41] and DNase-seq signals were from GEO: GSE18927.[42] The DNA sequence in the reference genome and genetic variants around the eQTL were shown in the upper right, and the YY1 binding motif from JASPAR was also shown.

(E) Scatterplots showing the ground truth of log₂-transformed fold change of STARR-seq signals upon EP300 perturbation (GEO: GSE156741[43]) and predictions by ChromBERT at test enhancers. The Pearson's correlation coefficient is annotated in the plot, and the color represents the density of points.

(F) Heatmap showing the shifts in regulator embeddings of potential cofactors of EP300 at the test enhancers before and after perturbation for EP300/CREBBP. Two groups of enhancers were depicted: those with decreased activity (log₂-transformed fold change < −1, $n$ = 164) and those unchanged (−0.5 < log₂-transformed fold change < 0.5, $n$ = 343) after IAA-treated EP300/CREBBP depletion. The color represents the shifts (1 − cosine similarity) in embeddings. Representative regulators showing high embedding shift at decreased enhancers are highlighted in blue, and representative regulators showing low embedding shift at decreased enhancers are highlighted in red.

(G) Line charts showing the average ChIP-seq profile of BRD4 around test enhancers. BRD4 ChIP-seq data were from a previous study (GEO: GSE57628[44]).

(H) Heatmap showing log₂-transformed fold changes of STARR-seq signals after IAA-treated EP300/CREBBP or BRD4 depletion; test enhancers were divided into two groups with decreased or unchanged activity after IAA-treated EP300/CREBBP depletion.

(I) Circos plot showing embedding similarities with EZH2 at classical (w/ H3K27me3) and non-classical (w/o H3K27me3) genomic loci, highlighting two groups of regulators having locus-preferential embedding similarities with EZH2. The classical group shows higher embedding similarity with EZH2 at classical loci compared to non-classical loci (embedding similarity with EZH2 at classical loci ranking the top 5% among all regulators and the embedding similarity difference between classical loci and non-classical loci > 0.1), and the non-classical group exhibits the converse pattern. The classical group was marked in green, and the non-classical group was marked in yellow. The known cofactors in the classical function of EZH2 were highlighted in blue, and the known cofactors in the non-classical function of EZH2, E2F1, and STAT3 were highlighted in red. Each node represents a regulator, and the transparency of edges linking two nodes represents the embedding similarity of two regulators; only edges with high pairwise embedding similarity (> 0.8) were plotted. See also Figures S6–S9.

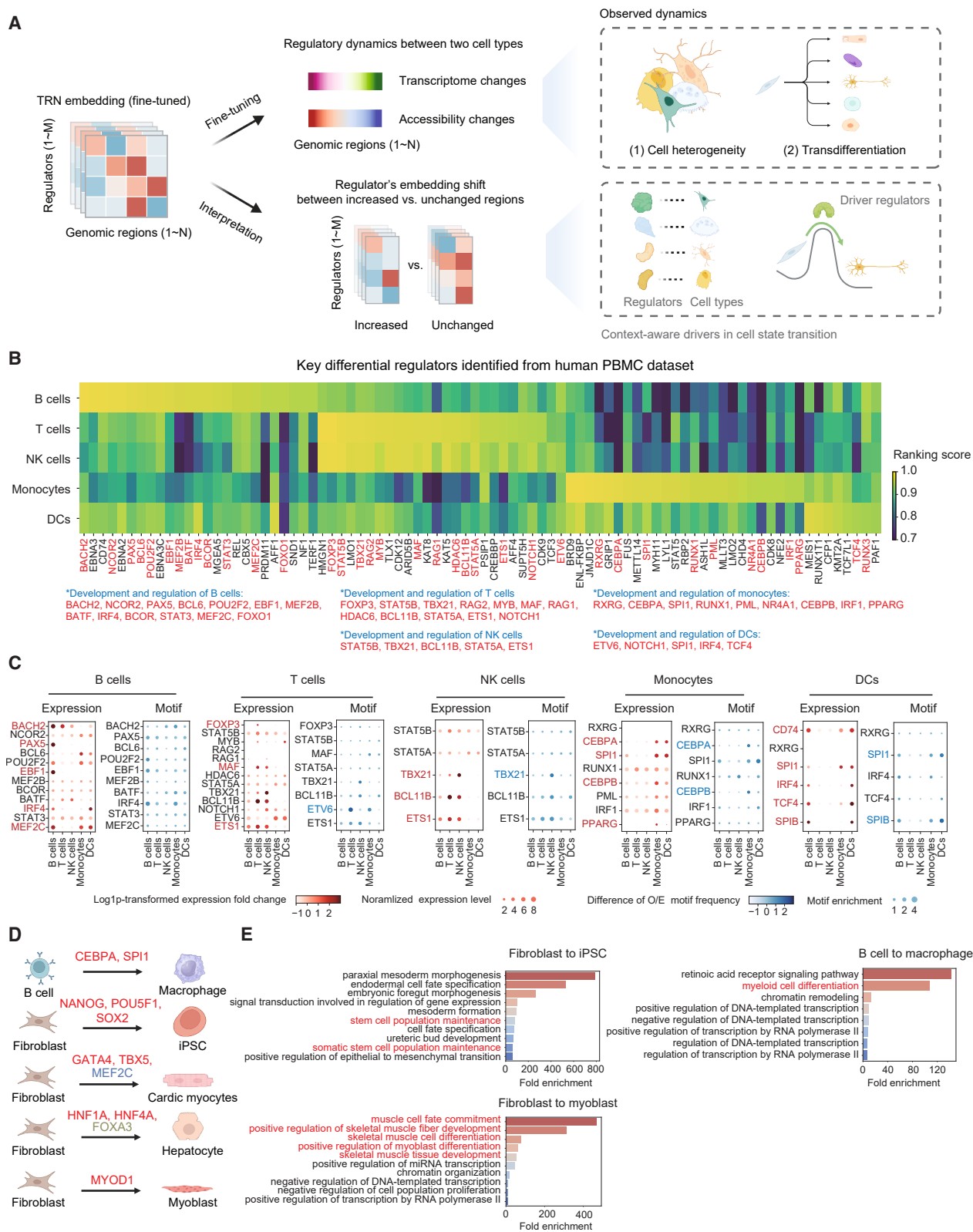

demonstrate the ability of ChromBERT to decipher the genomic-context-dependent cooperation of transcription regulators.

## ChromBERT uncovers key drivers in cell state transition

Understanding the transcriptional regulatory dynamics underlying cell state transition is crucial for gaining deep insights into development, disease processes, and drug discovery. To investigate these dynamics and identify key regulators involved in cell state transitions, we fine-tuned ChromBERT to predict measured genome-wide changes in gene expression or chromatin accessibility between pairs of cell types. As in interpretation for cell-type-specific regulatory effects, we then interpreted the task-adapted, context-specific embeddings by comparing genomic intervals (or genes) that exhibit changes with those that do not. We then ranked regulators based on the magnitude of their embedding shifts to identify those most likely to drive the transition (Figure 4A; see STAR Methods).

We first analyzed the transcriptional regulatory dynamics within the human PBMC single-cell multi-omics dataset that mainly comprises five cell types: B cells, T cells, natural killer (NK) cells, dendritic cells (DCs), and monocytes, defined by marker genes. ChromBERT was fine-tuned to predict the genome-wide changes in the transcriptome or chromatin accessibility between these cell types in a pairwise manner (see STAR Methods), and it outperformed all other methods (Figure S10). To evaluate the impact of each transcription regulator, we analyzed the embedding shift between two groups of genomic regions characterized by increased versus unchanged expression or chromatin accessibility in the two cell types. This interpretation highlighted the regulators that drive cell heterogeneity and pinpointed regulators unique to each cell type (see STAR Methods). Notably, 37 out of 84 identified key regulators have previously been reported to play important functions in the corresponding cell types (Figure 4B). For instance, BACH2, NCOR2, PAX5, BCL6, BCOR, and IRF4 are well-characterized factors involved in the transcription regulation of memory B cell differentiation.[57] Similarly, FOXP3 and TBX21 are known regulators in T cell lineage commitment,[58,59] and CEBPA, CEBPB, and SPI1 are well-characterized regulators in monocytes.[60–62] Strikingly, many of these known functional regulators recapitulated by ChromBERT could not be detected through differential expression analysis or motif enrichment analysis in differentially accessible chromatin (Figure 4C), two common approaches for key regulator discovery.[63] Additionally, SCENIC+, a computational framework for single-cell multi-omics inference of gene regulatory networks,[64] also failed to predict most known functional regulators in this dataset (Figure S11A). Remarkably, the fine-tuned ChromBERT assigned these factors larger embedding shifts and higher rankings than the pre-trained model (Figures S11B and S11C), confirming the effectiveness of fine-tuning and context-specific representation adaptation in uncovering important regulators in cell heterogeneity. These findings suggest that ChromBERT can accurately capture the dynamics of TRNs across heterogeneous cell types and highlight important transcription regulators.

Next, we focused on transdifferentiation, in which one mature cell type is directly reprogrammed into another. ChromBERT was also fine-tuned on genome-wide changes in gene expression or chromatin accessibility across five transdifferentiation paradigms and again outperformed all other methods (Figures S12A and S12B). To uncover driver regulators, we ranked each factor's embedding shift between genomic regions that gained versus retained expression or accessibility during the transition (see STAR Methods). This interpretation analysis recovered nearly all known drivers of the five transitions (Figures 4D and S12C). The remaining candidates showed higher embedding similarity with these known driver regulators (Figure S12D), suggesting that newly identified transcription regulators may function collaboratively with known drivers. For instance, identified key regulators in three cell state transition processes, from fibroblast to induced pluripotent stem cells, from B cells to macrophage, and from fibroblast to myoblast,

**Figure 4. ChromBERT uncovers key regulators in cell state transition**

(A) Schematic of the fine-tuning of ChromBERT in modeling regulatory dynamics between two cell types and interpreting the underlying roles of transcription regulators. From left to upper right, ChromBERT is fine-tuned to predict observed regulatory dynamics between cell types, where the task objective is genome-wide changes in the transcriptome or chromatin accessibility. From left to lower right, fine-tuned TRN embeddings are interpreted to reveal context-specific roles of transcription regulators corresponding to the observed regulatory dynamics during cell state transition (see STAR Methods). The key principle in interpretation is that when a regulator's embedding shifts systematically between regions with increased versus unchanged expression or chromatin accessibility, the regulators showing the largest shifts are inferred to drive the change in the given cell state transition (Figure 1B). This schematic was created in BioRender. Yu, Z. (2025) https://BioRender.com/3biab7r.

(B) Heatmap showing cell-type-specific ranking scores of all key regulators identified in five major cell types: B cells, T cells, NK cells, monocytes, and DCs. The cell-type-specific ranking scores evaluate the importance of regulators in differentiating the given cell type from all other cell types (see STAR Methods). All key regulators identified were generally classified into four categories: B cell-specific, T cell/NK cell-specific, monocyte-specific, and DC-specific, according to the cell-type-specific ranking scores for all 84 identified key regulators. Known driver regulators in each cell type were highlighted in red and specifically annotated below the heatmap.

(C) Dot plots show which known drivers were recovered by (1) differential expression (red dots) or (2) differential motif enrichment (blue dots). A regulator was defined as differentially expressed if its log1p-transformed expression level in the given cell type exceeds the mean in others by > 1; it was defined as differential motif enrichment if its motif's observed/expected frequency in accessible regions exceeds the mean in others by > 1. Dot size encodes expression or motif-enrichment magnitude; color shows the difference versus other cell types. Log1p refers to log(x+1) calculation.

(D) The extent to which ChromBERT recapitulates known driver regulators across five transdifferentiation processes. Known driver regulators recapitulated by ChromBERT were highlighted in red. MEF2C, not recapitulated by ChromBERT, was marked in blue. FOXA3, lacking related cistromes in our model, was marked in brown.

(E) GO enrichment analysis for 25 identified key regulators using DAVID online tool (https://david.ncifcrf.gov/tools.jsp).[56] The top ten human GO biological process terms with a false discovery rate < 0.05 and high fold enrichment were displayed. The GO terms involved in the terminal cell states were highlighted in red. See also Figures S10–S12.

were highly enriched in gene ontology (GO) terms related to the target cell fate determination (Figure 4E). As in the PBMC analysis, known drivers received higher ranks and exhibited larger embedding shifts in the fine-tuned model than in the pre-trained model (Figures S12E and S12F), underscoring the significance of task-specific adaptation and context-specific interpretability. We further validated the embedding-shift analysis using perturbation datasets. Despite the subtle differences between perturbation and control conditions, ChromBERT correctly identified the perturbed regulator in 3 out of 5 tested cases using a recently published ATAC-seq dataset with transcription regulator perturbations[65] (Figure S12G). These results demonstrate ChromBERT's ability to identify key transcription regulators driving transdifferentiation processes and cell state transitions, providing valuable insights for targeted cell-fate engineering.

## DISCUSSION

ChromBERT's modular architecture and transfer learning scheme create numerous opportunities for new applications in transcription regulation research. In single-cell omics, fine-tuning the model on multi-omics profiles could reveal how TRNs shift across heterogeneous cell populations and over developmental time or in response to perturbations. Extending ChromBERT to model context-dependent interactions across multiple genomic windows would further enable the identification of enhancer-promoter pairs and higher-order chromatin contacts from TRNs, bridging local regulatory inference with 3D genome organization. Prompt-enhanced fine-tuning also allows ChromBERT to work synergistically with other foundation models. Integrating ChromBERT's TRN embeddings with DNA sequence representations from sequence-based foundation models could jointly capture sequence grammar and regulatory context, potentially supporting cross-species inference of regulatory networks through conserved genomic elements. Similarly, combining ChromBERT with single-cell foundation models could yield insights into the gene-regulatory mechanisms underlying dynamic cell states and transitions, offering deeper insights into cellular plasticity and lineage specification.

While the present study primarily focused on predictive accuracy and key regulator identification through embedding similarity and shift analyses, ChromBERT provides a promising foundation for explicit reconstruction of TRNs within specific cellular contexts. Each TRN embedding captures contextual dependencies among transcription regulators within a genomic region via the self-attention mechanism, which encodes both pairwise and higher-order regulatory interactions. By aggregating these contextual relationships, derived from embedding similarity matrices or attention-weight distributions, across biologically meaningful genomic regions, ChromBERT could systematically reconstruct accurate and stable hierarchical TRN structures that reflect global co-regulatory patterns. Furthermore, by fine-tuning ChromBERT using chromatin accessibility or transcriptomic profiles from specific cell types, the TRN embeddings become adapted to the corresponding cellular context. This adaptation would refine the learned regulator-regulator interaction structure to reflect cell-type-specific regulatory logic. Such reconstructions would facilitate comparative analyses of regula-

tory network rewiring across development, differentiation, or disease progression. Edges within the regulatory network could be further refined or weighted with external evidence to strengthen its biological interpretability and directionality, for example: (1) motif constraint, retaining only edges where the source regulator has a known binding motif near the target's regulatory sites; (2) ChIP-seq or ATAC-seq support, retaining only edges supported by experimental binding or accessibility evidence; and (3) perturbation-based functional screens, validating that perturbing the source regulator alters the activity of the target, which also informs edge directionality.

## Limitations

Despite its promise, ChromBERT still faces several limitations that restrict its utility for modeling transcriptional regulation. First, its pre-training corpus covers cistromes for only approximately one thousand transcription regulators. Many regulators lack public ChIP-seq data, so they remain unrepresented. Moreover, because cell-type-specific cistromes are scarce, existing tracks must be pooled across cell types. This broadens coverage but blurs cell-type specificity. Expanding ChIP-seq datasets and tagging each experiment with explicit cell-type metadata would let future versions of ChromBERT be pre-trained in a cell-type-specific manner. Second, aligning ChromBERT with other foundation models is challenging because each modality, DNA sequence, 3D chromatin structure, and single-cell transcriptomics, resides in its own latent space. A long-term aim is to pre-train a unified, multimodal foundation model from scratch, jointly ingesting DNA sequence, long-range chromatin contacts, gene expression profiles, and cistromes. Such an integrative model would place all modalities in a common embedding space, eliminate post hoc alignment issues, and provide a comprehensive framework for decoding transcriptional regulation. Finally, certain hyperparameters in the current pre-trained framework, such as the number of transformer blocks and hidden dimension sizes were not systematically optimized. Conducting a thorough hyperparameter search tailored to genomic inputs may further improve the model's performance.

## RESOURCE AVAILABILITY

### Lead contact

Requests for further information and resources should be directed to and will be fulfilled by the lead contact, Yong Zhang (yzhang@tongji.edu.cn).

### Materials availability

This study did not generate new, unique reagents.

### Data and code availability

Cistrome-Human-6K dataset is available on the Hugging Face Dataset Hub (https://huggingface.co/datasets/TongjiZhanglab/chrombert). The blacklist of genomic regions was obtained from the ENCODE project (https://www.encodeproject.org) under accession code ENCFF356LFX. The public data used in the model development and data analysis are listed in STAR Methods.

The pre-trained model (including pre-trained models in 200 bp, 1/2/4 kb resolution for humans, and pre-trained model in 1 kb resolution for mouse), reference datasets and all source codes of ChromBERT are available on the GitHub repository (https://github.com/TongjiZhanglab/ChromBERT) and Zenodo repository (https://doi.org/10.5281/zenodo.17824395).

## ACKNOWLEDGMENTS

We would like to thank Prof. Rui Jiang for comments on designing the research. This work was primarily supported by the National Natural Science Foundation of China (32325012 [Yong Zhang]) and the Science and Technology Commission of Shanghai Municipality (23JS1401200 [Yong Zhang]). This work was also supported by the National Natural Science Foundation of China (32030022 [Yong Zhang], 32488101 [Yong Zhang], and 32400522 [Z.Y.]), the National Key Research and Development Program of China (2021YFA1302500 [Yong Zhang]), the Postdoctoral Innovation Talents Support Program (BX20230265 [Z.Y.]), the China Postdoctoral Science Foundation (2022M722423 [Z.Y.]), and the GHfund C (202302033256 [Z.Y.]).

## AUTHOR CONTRIBUTIONS

Yong Zhang conceived the research; Yong Zhang and Z.Y. designed the research; Z.Y. and D.Y. developed the model; Q.C. and Yuxuan Zhang performed computational analysis with the help of Y.W. and Z.L.; and C.W., Z.Y., D.Y., Q.C., Yuxuan Zhang, and Yong Zhang wrote the manuscript.

## DECLARATION OF INTERESTS

The authors declare no competing interests.

## STAR★METHODS

Detailed methods are provided in the online version of this paper and include the following:

- KEY RESOURCES TABLE
- METHOD DETAILS
  - Assembly and binding status encoding of cistromes
  - Architecture and pre-training of ChromBERT
  - Representations of ChromBERT
  - Perturbation and omission of cistromes
  - Fine-tuning of ChromBERT
  - Downstream tasks
  - Benchmarks
  - Gene ontology enrichment analysis
  - Motif scan
- QUANTIFICATION AND STATISTICAL ANALYSIS

## SUPPLEMENTAL INFORMATION

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

# STAR★METHODS

## KEY RESOURCES TABLE

| REAGENT or RESOURCE | SOURCE | IDENTIFIER |
|---|---|---|
| **Deposited data** | | |
| Cistrome-Human-6K dataset | This paper | https://huggingface.co/datasets/TongjiZhanglab/chrombert |
| Blacklist of genomic regions | ENCODE | ENCODE: ENCFF356LFX |
| Hi-C data for K562 | Gene Expression Omnibus | GEO: GSE63525 |
| Hi-C data for hESC | 4D Nucleome | 4D Nucleome: 4DNES2M5JIGV |
| Hi-C data for HFF | 4D Nucleome | 4D Nucleome: 4DNES2R6PUEK |
| Micro-C data for hESC | 4D Nucleome | 4D Nucleome: 4DNES21D8SP8 |
| Micro-C data for HFF | 4D Nucleome | 4D Nucleome: 4DNESWST3UBH |
| ChIP-seq data used as ground truth for cistrome imputation task | ENCODE and Gene Expression Omnibus (see Tables S1 and S2) | Tables S1 and S2 |
| DNase-seq data used as prompt for cistrome imputation task | ENCODE and Gene Expression Omnibus (see Tables S1 and S2) | Tables S1 and S2 |
| RNA-seq data used as prompt for cistrome imputation task | ENCODE and Gene Expression Omnibus (see Table S1 and S2) | Table S1 and S2 |
| ChIP-seq data used for cistrome imputation task: BRD4 of A549 | Gene Expression Omnibus | GEO: GSE89128 |
| ChIP-seq data used for cistrome imputation task: BRD4 of K562 | Gene Expression Omnibus | GEO: GSE99178 |
| ChIP-seq data used for cistrome imputation task: EP300 of HCT116 | Gene Expression Omnibus | GEO: GSE51176 |
| ChIP-seq data used for cistrome imputation task: EP300 of IMR-90 | Gene Expression Omnibus | GEO: GSE59681 |
| DNase-seq data used for cistrome imputation task: A549 | Gene Expression Omnibus | GEO: GSE26328 |
| DNase-seq data used for cistrome imputation task: K562 | Gene Expression Omnibus | GEO: GSE25344 |
| DNase-seq data used for cistrome imputation task: HCT116 | Gene Expression Omnibus | GEO: GSE50610 |
| DNase-seq data used for cistrome imputation task: IMR-90 | Gene Expression Omnibus | GEO: GSM723024 |
| PBMC 10× single-cell multi-omics dataset | 10× Genomics Datasets | https://www.10xgenomics.com/datasets/pbmc-from-a-healthy-donor-granulocytes-removed-through-cell-sorting-10-k-1-standard-2-0-0 |
| Annotations for eQTLs | EBI eQTL catalog | https://www.ebi.ac.uk/eqtl |
| ChIP-seq data used for eQTL identification: YY1 in lung | Gene Expression Omnibus | GEO: GSE32465 |
| DNase-seq data used for eQTL identification: lung | Gene Expression Omnibus | GEO: GSE18927 |
| DNase-seq data for HCT116 | ENCODE | ENCODE: ENCFF240LRP |
| List of enhancer regions for analysis of STARR-seq | Neumayr et al.[43] | https://doi.org/10.1038/s41586-022-04779-x |
| STARR-seq data | Gene Expression Omnibus | GEO: GSE156740 |
| ChIP-seq data for BRD4 of HCT116 | Gene Expression Omnibus | GEO: GSE57628 |
| ChIP-seq data for H3K27me3 in hESC (used for RNF2 dual roles analysis) | Gene Expression Omnibus | GEO: GSE62562 |
| ChIP-seq data for H3K27me3 in hESC (used for EZH2 dual roles analysis) | Gene Expression Omnibus | GEO: GSE61176 |
| ChIP-seq data for EZH2 | Gene Expression Omnibus | GEO: GSE29611 |
| DNase-seq data used for modeling the chromatin accessibility changes: fibroblast | ENCODE | ENCODE: ENCFF184KAM |

*(Continued on next page)*

*Continued*

| REAGENT or RESOURCE | SOURCE | IDENTIFIER |
|---|---|---|
| DNase-seq data used for modeling the chromatin accessibility changes: iPSC | ENCODE | ENCODE: ENCFF540VPT |
| DNase-seq data used for modeling the chromatin accessibility changes: myoblast | ENCODE | ENCODE: ENCFF647RNC |
| DNase-seq data used for modeling the chromatin accessibility changes: cardiac myocyte | ENCODE | ENCODE: ENCFF389SOW |
| DNase-seq data used for modeling the chromatin accessibility changes: B cell | ENCODE | ENCODE: ENCFF650BNV |
| DNase-seq data used for modeling the chromatin accessibility changes: macrophage | ENCODE | ENCODE: ENCFF580ICE |
| ATAC-seq data used for modeling the chromatin accessibility changes: fibroblast | Gene Expression Omnibus | GEO: GSE179011 |
| ATAC-seq data used for modeling the chromatin accessibility changes: hepatocyte system | Gene Expression Omnibus | GEO: GSE185358 |
| CAGE-seq data used for modeling the transcriptome changes | FANTOM5 | https://fantom.gsc.riken.jp/5/datafiles/ reprocessed/hg38_latest/extra/CAGE_peaks_ expression/hg38_fair+new_CAGE_peaks_ phase1and2_tpm.osc.txt.gz |
| **Software and algorithms** | | |
| ChromBERT | This paper | https://github.com/TongjiZhanglab/ChromBERT (Zenodo: https://doi.org/10.5281/zenodo. 17824395) |
| Bedtools (v2.30.0) | Quinlan et al.[66] | https://github.com/arq5x/bedtools2 |
| FIMO (v5.0.5) | Grant et al.[40] | https://meme-suite.org/meme/ |
| Flash-attention (v2.3.2) | Tri et al.[67] | https://github.com/Dao-AILab/flash-attention |
| MUON (v0.1.5) | Danila et al.[68] | https://github.com/scverse/muon |
| liftOver (latest) | UCSC Genome Browser team | https://hgdownload.soe.ucsc.edu/admin/exe/ linux.x86_64/liftOver |
| pycisTopic (v1.0.3) | Bravo et al.[64] | https://github.com/aertslab/pycisTopic |
| pytorch (v2.0.1) | PyTorch | https://pytorch.org |
| pytorch-lightning (v2.0.4) | Lightning AI | https://github.com/Lightning-AI/pytorch-lightning |
| Scanpy (v1.9.5) | Wolf et al.[69] | https://github.com/scverse/scanpy |
| transformers (v4.28.1) | HuggingFace | https://github.com/huggingface/transformers |

## METHOD DETAILS

### Assembly and binding status encoding of cistromes
#### Assembly of cistromes in Cistrome-Human-6K

To create a comprehensive pre-training corpus, we gathered data from a wide range of cell types using publicly available sources. Initially, we collected all human ChIP-seq, DNase-seq, and ATAC-seq data (cistromes) from the Cistrome Data Browser.[24] We then applied several filtration steps to select qualified datasets.

- Quality control metrics: We included datasets that met at least four out of five quality control metrics provided by the Cistrome Data Browser: sequence median quality score ≥25, uniquely mapped ratio ≥60%, PCR bottleneck coefficient ≥80%, fraction of reads in peaks ≥1% and number of 10-fold confident peaks ≥500.
- Peak counts: We recorded datasets with more than 100 called peaks for transcription factors and over 1000 called peaks for other transcription regulators.
- Genome coverage: We ensured that more than 60% of genome-wide regions without called peaks were covered by reads.
- Replicate removal: To ensure accuracy in our annotations, we followed a structured approach to associate datasets with cell populations. Specifically, when a dataset explicitly included a cell line annotation, we used the cell line as the primary label. For datasets without cell line annotations, we assigned labels based on a combination of cell type and tissue to avoid conflating datasets with similar cell types derived from different tissues. This approach ensures consistent and biologically meaningful categorization. To avoid redundancy, we retained the highest-ranking dataset for each transcription regulator in the same

cell line/cell type, based on the combined score of all quality control metrics. Given the inherent heterogeneity of tissues, all datasets derived from tissues that met our quality control criteria were retained for analysis.

Following these filtration steps, we assembled a dataset Cistrome-Human-6K consisting of 6,391 ChIP-seq, DNase-seq and ATAC-seq cistromes. Additionally, we manually curated its metadata to ensure its accuracy (Table S1). The dataset was subsequently processed into a signal matrix, $S \in \mathbb{R}^{N \times P}$, where each element $S_{i,j}$ represents average signal (reads per million mapped reads, RPM) of cistromes $c_j \in \{c_1, c_2, \cdots, c_P\}$ ($P = 6,391$) at the $i$-th bin $b_i \in \{b_1, b_2, \cdots, b_N\}$ in the genome. The set $\{b_1, b_2, \cdots, b_N\}$ denotes consecutive 1-kb bins split from the human build genome hg38. To avoid potential sequencing bias, blacklist regions from ENCODE were excluded from the analysis (https://www.encodeproject.org/files/ENCFF356LFX/). To focus on co-binding events at genomic regions, only bins with peak presence in more than two cistromes were kept in the analysis. Ultimately, a total of 2,137,894 1-kb bins were retained in the matrix.

### Binding status encoding of cistromes

Transcription regulators often co-bind to chromatin and function in a combinatorial manner.[1] In ChromBERT, we model the genome-wide combinatorial binding patterns of cistromes to generate representations for TRNs. Therefore, the input into ChromBERT is the binding statuses of all assembled cistromes at each individual 1-kb bin $b_i$, which were processed from signal vector $S_i = [S_{i,1}, S_{i,2}, \cdots, S_{i,P}]$ at genomic bin $b_i$.

To ensure the comparability and suitability of signals from different cistromes for input into the deep language model, we first discretized these signals into distinct categories representing different binding statuses. Specifically, for each cistrome $c_j$, we categorized its signals across the entire genome into two groups, "binding" or "non-binding". The threshold was defined as the 10-th percentile of signals from 1-kb bins with peak presence. Peak files were downloaded from Cistrome Data Browser.[24] This categorization is designed to highlight the biologically meaningful but sparse binding events of transcription regulators in the genome. The "binding" category was further divided into three equal portions, "slightly positive", "moderately positive" and "strongly positive", based on the signals within this group. Similarly, the "non-binding" group was divided into two equal portions, "slightly negative" and "strongly negative" (Figure S1A). This finer classification aims to incorporate the impact of binding strength in modeling the co-association pattern of transcription regulators. After binding status categorization for all cistromes, the signal vector can be converted to a binding status vector $X_i = [X_{i,1}, X_{i,2}, \cdots, X_{i,P}]$, where $X_{i,j}$ corresponding to the defined five categories of binding statuses from "strongly negative", "slightly negative", "slightly positive", "moderately positive" to "strongly positive".

### Architecture and pre-training of ChromBERT
### The architecture of ChromBERT

ChromBERT architecture includes three parts: (1) an input embedding layer to generate input embeddings from input features; (2) eight transformer blocks (encoder); and (3) output heads adapted to different tasks (decoder).

The input embedding layer converts the binding status vector of each 1-kb bin into a high-dimension representation. We chose a 1-kb window size because most transcription regulator co-binding events occur within a few hundred base pair. A 1-kb bin has been widely adopted in genome-wide studies of transcriptional regulation, as it provides an appropriate resolution for capturing local cooperative or competitive interactions among regulators. Nevertheless, we also pre-trained ChromBERT models using 200 bp, 2 kb, and 4 kb windows (see Code availability). The binding status vector in each 1-kb bin, $X_i$, is analogous to a sentence in natural language processing, where binding status of each cistrome, $X_{i,j}$, is considered as a word in the sentence. Therefore, we utilize the similar input representation approach in BERT model,[22] to generate input embeddings from $X_i$. The input embeddings include position embeddings that represent different cistromes and token embeddings that represent different binding statuses. Each word in $X_i$ is assigned a unique integer position identifier to distinguish individual cistromes, resulting in a position vector for all words, denoted as $v = [id(1), id(2), \cdots, id(P)]$. And token vector is $X_i$, where each item $X_{i,j}$ belongs to one of five binding status categories. Then ChromBERT employ the commonly used embedding layers in PyTorch (v2.0.1),[70] $Emb_{position}$ and $Emb_{token}$, for two vectors respectively, to map each position and token to a fixed-length embedding vector of dimension $d_{input} = 768$. Consequently, the resultant summarized input embeddings for each bin $b_i$, $E_i^{(input)} \in \mathbb{R}^{P \times d_{input}}$, is defined as:

$$E_i^{(input)} = Emb_{position}(v) + Emb_{token}(X_i) \qquad \text{(Equation 1)}$$

$E_i^{(input)}$ represents the binding statuses of $P$ cistromes at the same genomic bin $b_i$, facilitating the following modeling of their co-association patterns with transformers.

ChromBERT employs a transformer encoder to convert input embeddings $E_i^{(input)}$ into TRN embeddings $E_i^{(TRN)} \in \mathbb{R}^{P \times d_{model}}$, which represents context-specific TRN, where $d_{model} = 768$. We adopted a 768-dimensional hidden dimension for ChromBERT based on the widely used BERT-base architecture in natural language processing (NLP),[22] which provides a well-balanced trade-off between computational efficiency and representational capacity. The encoder comprises eight self-attention transformer blocks,[22,23] each consisting of a multi-head self-attention layer and a feedforward neural network layer (Figure S1B). These self-attention mechanisms operate on the input embeddings, allowing the model to capture intricate interactions among all cistromes. The self-attention mechanism of $l$-th block operates as follows:

$$Q = E_i^{(l-1)} W_q^{(l)}, K = E_i^{(l-1)} W_k^{(l)}, V = E_i^{(l-1)} W_v^{(l)} \qquad \text{(Equation 2)}$$

$$Attention(Q, K, V) \; = \; softmax\left(\frac{QK^T}{\sqrt{d_{model}}}\right)V \tag{Equation 3}$$

Here, $E_i^{(l-1)}$ is the output of the $(L$-1)-th block, and $E_i^{(0)}$ is $E_i^{(input)}$, $Q, K$ and $V$ are the query, key and value vectors respectively, each with dimension $P \times d_{model}$. The weight matrices $W_q^{(l)}, W_k^{(l)}, W_v^{(l)}$ are of dimensions $d_{model} \times d_{model}$, and $W_q^{(0)}, W_k^{(0)}, W_v^{(0)}$ are of dimensions $d_{input} \times d_{model}$. Additionally, ChromBERT splits the computation of self-attention into eight attention heads to allow the model to jointly attend to information from different representation subspaces at different positions. To enhance processing efficiency for the large sequence length ($P$ = 6,391), ChromBERT incorporates FlashAttention-2 (v2.3.2)[67], which optimizes attention computation through improved parallelism and work partitioning.

Due to the lack of explicit ground truth for genome-wide combinatorial interactions among transcription regulators, ChromBERT's pre-training is performed in a self-supervised manner. This involves a decoder architecture that follows the encoder to reconstruct input binding status vector $X_i$ from the TRN embedding $E_i^{(TRN)} \in \mathbb{R}^{P \times d_{model}}$, producing an output $\widehat{Y}_i \in \mathbb{R}^{P \times 5}$. The decoder is a two-layer multilayer perceptron (MLP): the first MLP layer applies a linear transformation with parameters of shape ($d_{model}, d_{model}$), followed by a ReLU activation. The second MLP layer then projects the hidden dimension to the output vocabulary dimension with parameters of shape ($d_{model}$, 5).The model setup and training processes are facilitated using PyTorch, ensuring robust configuration, data handling, and computational efficiency.

### Pre-training with masked learning
ChromBERT employs a masked learning approach, widely used in pre-trained models, to enhance the generalizability of fundamental knowledge acquired during pre-training, which allows for efficient and accurate transfer learning in a variety of downstream fine-tuning tasks.[5,6,8,22,71] In the pre-training of ChromBERT, 15% of the binding statuses in the input sequence $X_i$ were randomly selected. The model's pre-training objective is to predict these masked statuses based on the context provided by the remaining cistromes. For these 15% binding statuses, we applied a similar masked learning strategy used in BERT.[22] In details, we replaced the binding statuses $X_{i,j}$ with (1) the [MASK] in 80% of the time or (2) a random binding status in 10% of the time or (3) the unchanged $X_{i,j}$ in 10% of the time. Through this process, ChromBERT gained deep insights into the grammar that governs the combinatorial interactions among transcription regulators, effectively learning to infer masked elements.

Due to the severe class imbalance among the five binding status categories, focal loss[72] was employed to refine the training process. This loss function modifies the standard cross-entropy loss to focus more on difficult-to-classify instances by decreasing the weight of easily classified examples. In the pre-training stage, only masked objectives were taken account in the loss computation. Let $y_j = Y_{i,j}$ denote as the true binding status category of the cistrome $c_j$ at the $i$-th bin $b_i$, then the focal loss function is defined as:

$$FL_{multi}(\widehat{y}, y) \; = \; -\sum_{c_j \in c_{masked}} \sum_{t=1}^{5} \left(1 - \widehat{y}_{j,t}\right)^{\gamma} \cdot y_{j,t} \cdot log\left(\widehat{y}_{j,t}\right) \tag{Equation 4}$$

In this formula, $\gamma$ is a focusing parameter, set to 2 by default, which scales the contribution of each example based on the ease of its classification. $y_{j,t}$ is an indicator variable equals to 1 if $X_{i,j} = t$, and equals to 0 for otherwise, where $t$ is one of the five binding status categories. $\widehat{y}_{j,t}$ is the predicted probability for the binding status cistrome $c_j$ being $t$. This adjustment ensures that the model pays greater attention to learning from challenging cases, thereby addressing the imbalance in the training data effectively.

Specific hyperparameters during the pre-training of ChromBERT were empirically selected as follows. The learning rate scheduler was configured to a linear schedule, incorporating a warm-up period that comprised 10% of all learning steps, increasing from 0 to the maximum learning rate $1 \times 10^{-4}$, and a decayed period that comprised remain 90%, decreasing from the maximum learning rate $1 \times 10^{-4}$ to 0. The AdamW optimizer was employed with its default settings to facilitate efficient optimization. The training batch size was set to 32, with gradient accumulation occurring every eight batches to manage memory effectively while achieving more stable gradient estimates. To speed up the training process, PyTorch Lightning (v2.0.4) (https://lightning.ai/pytorch-lightning) was used, enabling distributed computation across multiple GPUs. This framework significantly enhanced the training efficiency by parallelizing computations. ChromBERT underwent a rigorous training stage, spanning 100 epochs, which was completed in approximately 23 days with four NVIDIA A800 80GB GPUs. This intensive training period was necessary to ensure that the model adequately learned the complex patterns in the training data, setting a foundation for the subsequent fine-tuning stage.

### Representations of ChromBERT
#### TRN embeddings
ChromBERT is designed to represent TRNs across the genome by modeling the genome-wide combinatorial binding patterns of transcription regulators. For each genomic region $b_i$, ChromBERT encodes it into TRN embeddings $E_i^{(TRN)} \in \mathbb{R}^{P \times d_{model}}$. These embeddings are generated through transformer blocks, which provide a flexible architecture for fine-tuning the model's parameters to adapt to diverse downstream tasks. During pre-training, where randomly masked cistromes serves as learning objectives, the embeddings capture a general, cell-type-agnostic view of the TRNs. Fine-tuning on objectives from a particular cell type then adjust the embeddings to yield task-adaptive, context-specific TRN representations within the given cellular contexts. These embeddings offer crucial insights and context-specific biological interpretations, underpinning the model's utility in various transcription regulation-related applications.

Specifically, the $E_i^{(TRN)}$ focused on the interactions of $P$(6,391) cistromes at each genomic region, which are pooled ChIP-seq profiles of $M$(1,073) regulators. To improve the model's utility and interpretability, we transformed the cistrome-based TRN embeddings $E_i^{(TRN)}$ to regulator-based TRN embeddings $E_i'^{(TRN)} \in \mathbb{R}^{M \times d_{model}}$. This transformation involves characterizing the role of individual transcription regulators within the TRN. For regulators associated with multiple cistromes, we computed the average of all cistromes embeddings linked to a single regulator $r_m$ across different cell types at genomic bin $b_i$, resulting in $E_{i,m}'^{(TRN)} \in \mathbb{R}^{d_{model}}$, which is defined as follows:

$$E_{i,m}'^{(TRN)} = \frac{1}{|c_{r_m}|}\sum_{c_j \in c_{r_m}} E_{i,j}^{(TRN)} \qquad \text{(Equation 5)}$$

where $c_{r_m}$ represent all cistromes associated with $r_m$.

Finally, for each genomic bin $b_i$, ChromBERT generates a TRN embedding matrix $E_i'^{(TRN)} \in \mathbb{R}^{M \times d_{model}}$, in which each row of this matrix corresponds to a regulator embedding vector encoding the context-dependent regulatory role of that regulator after integrating information from all other regulators via the self-attention mechanism. These regulator-based embeddings were consistently applied in subsequent fine-tuning tasks and interpretability analysis by default. Notably, in the region-level analysis shown in Figure S2, which examines interactions among genomic regions, we computed an averaged TRN embedding vector $\overline{E_i^{(TRN)}} \in \mathbb{R}^{d_{model}}$ by averaging across all regulators, providing a compact representation for simplified visualization and quantitative analysis.

### Regulator embeddings
After transformation from cistrome-based TRN embedding to regulator-based TRN embedding, ChromBERT can output regulator embedding $E_{i,m}'^{(TRN)} \in \mathbb{R}^{d_{model}}$, for each regulator $r_m$ at genomic bin $b_i$.

### Inference for functional collaborations of transcription regulators
In assessing the functional collaborations of transcription regulators, we inferred their interactions based on the similarity of their regulator embeddings within TRN embeddings, derived from either a pre-trained model or a fine-tuned model achieving cell-type specificity. Specifically, for a given genomic region $b_i$, we computed the embedding similarity between pairs of regulators, denoted as $r_j$ and $r_k$. This was quantified using the cosine similarity formula:

$$CS_{b_i}(r_j, r_k) = \frac{E_{i,j}'^{(TRN)} \cdot E_{i,k}'^{(TRN)}}{\left\|E_{i,j}'^{(TRN)}\right\| \left\|E_{i,k}'^{(TRN)}\right\|} \qquad \text{(Equation 6)}$$

where $E_{i,j}'^{(TRN)}$ and $E_{i,k}'^{(TRN)}$ represents regulator embeddings for $r_j$ and $r_k$ at bin $b_i$ from the TRN embeddings $E_{i,j}'^{(TRN)}$. Furthermore, for functional collaborations across a specific set of bins, we averaged the regulator embeddings across all bin in the set. The embedding similarity was then calculated in a similar manner, enabling a broader understanding of regulator interactions across multiple genomic regions.

## Perturbation and omission of cistromes
### Perturbation of cistromes
ChromBERT takes the genome-wide binding events of around one thousand transcription regulators as input, which can be perturbed for modeling the impact of binding events of individual regulator on the model output. This involved modifying the binding events of cistromes associated with each transcription regulator $r_m$ across the genome. Specifically, we transformed the original input $X_{i,j}$ into a knockout input $X_{i,j}'$, where converted all categories with positive or "slightly negative" binding status to "strongly negative" category. For each perturbed regulator $r_m$, this perturbation process was applied to all associated cistromes $c_j \in c_{r_m}$.

### Omission of cistromes
Generally, ChromBERT processes binding status vector $X_i = [X_{i,1}, X_{i,2}, \cdots, X_{i,P}]$ ($P = 6,391$). However, in certain analyses, highly dominant cistromes may overshadow the contributions of cofactors related to the learning objectives. For example, the presence of H3K27me3 is strongly correlated with EZH2's classical role in gene expression repression. This dominance often leads to the exclusion of other regulators during fine-tuning tasks of classifying classical versus non-classical EZH2 binding sites. As a result, omitting H3K27me3 cistromes can enhance the identification of other key regulators. Formally, we yield a new binding status vector $X_i' = [X_{i,1}, X_{i,2}, \cdots, X_{i,P}]$ where $c_j \notin c_{r_m}$ by omitting all cistrome related with $r_m$.

## Fine-tuning of ChromBERT
### Basic principle of fine-tuning
The fine-tuning of ChromBERT was conducted using a selective training strategy to effectively leverage its pre-trained knowledge. Initially, ChromBERT was loaded with pre-trained weights to retain the general representations learned during pre-training. To preserve this foundational knowledge, the early layers of the encoder were frozen, while the final layers of the encoder and all layers of the task-specific decoder were made trainable. This approach ensures that ChromBERT retains its pre-trained knowledge while adapting effectively to the specific requirements of diverse downstream tasks. By selectively adjusting the model, ChromBERT can adapt its TRN embeddings to align with task-specific and cell-type-specific objectives. This capability not only enables the model to

perform well across a range of downstream tasks but also facilitates the context-specific interpretation of the roles and functional collaborations of transcription regulators in cell-type-specific transcription regulation.

### Fine-tuning for cell-type-specific regulatory effects

For each downstream task for cell-type-specific regulatory effects, we fine-tuned ChromBERT with region-level labels that capture the measured regulatory outcome in a given cell type (e.g., enhancer activity or gene expression level across the genome). The encoder first yields a TRN embedding for every 1-kb bin; a task-agnostic decoder, one convolution layer followed by a multi-layer perceptron (MLP), then maps that embedding to a single output node, returning a probability for classification tasks or a scalar for regression tasks. During this kind of fine-tuning tasks, we unfreeze the last $N$ transformer blocks, allowing the model to adjust regulator-to-regulator attention weights so that each embedding becomes context-specific. Notably, this design also supports tasks with inputs subjected to perturbation or omission of cistromes. Further details on this kind of downstream tasks were described in the following downstream tasks section and the corresponding training settings for fine-tuning (including learning rate, batch size, loss function, number of epochs, and number of frozen transformer blocks) were summarized in Table S3.

### Fine-tuning for regulatory dynamics between cell types

In our study, we utilized ChromBERT to understand transcriptional regulatory dynamics in single cell heterogeneity and during trans-differentiation. Specifically, the model was fine-tuned to predict genome-wide changes in transcriptome or chromatin accessibility between pairwise cell types. The fine-tuning process for chromatin accessibility followed the same objectives as those used for determining cell-type-specific regulatory effects. However, the objectives for transcriptome changes differed due to the potential uncertainty in the influence of regions adjacent to transcription start sites (TSSs) on gene expression. For each gene, we included both upstream and downstream regions relative to the TSS to capture a comprehensive genomic context. We selected four genomic bins on either side of the TSS, as well as the bin containing the TSS itself, denoted as $[b_{i-4}, b_{i-3}, b_{i-2}, b_{i-1}, b_i, b_{i+1}, b_{i+2}, b_{i+3}, b_{i+4}]$. To derive the most informative TRN embeddings from these regions, we then employed a max pooling strategy. This method ensures that the most significant signals across these genomic bins are captured, enhancing the accuracy of our predictions regarding gene expression regulation. A universal decoder architecture was followed to transform the TRN embeddings to the requisite output. Further details on this kind of downstream tasks were described in the following downstream tasks section and the corresponding training settings for fine-tuning (including learning rate, batch size, loss function, number of epochs, and number of frozen transformer blocks) were summarized in Table S3.

### Fine-tuning with prompts for additional information incorporation

Due to the mixed cell type pooling operation and reliance on cistromes involved in pre-training, the encoder of ChromBERT exhibits a limited capacity to generate variable TRN embeddings in response to specific inputs. In this study, we employed a prompt-enhanced fine-tuning approach, integrating TRN representations with additional relevant information, including cell-type-specific data or genetic variations. This approach successfully addresses these limitations and broadens the model's applicability across diverse objectives, including cistromes imputation and the identification of causal eQTLs. By annotating cistromes, cell-type-specific chromatin accessibility representation derived from pre-trained ChromBERT embedding can be utilized as cell-type-specific information for cistromes imputation. Additionally, transcriptome embeddings from scGPT were also employed as cell-type-specific information in this study. Genetic variation information we used was sourced from DNABERT-2 for identification of causal eQTLs. Vectors containing additional information were denoted as prompts in this study. To utilize the TRN representations and prompts, we concatenated each TRN embeddings $E_i'^{(TRN)}$ with respective prompts $E_i^{(prompts)}$. This concatenation produced a prompt-enhanced embeddings $E_i''^{(TRN)}$, which was subsequently processed by an MLP decoder to generate the necessary outputs. Further details on this kind of downstream tasks were described in the following downstream tasks section and the corresponding training settings for fine-tuning (including learning rate, batch size, loss function, number of epochs, and number of frozen transformer blocks) were summarized in Table S3.

### Interpretation analysis for fine-tuned regulator embeddings

In this study, we used embedding-based interpretation rather than attention matrices-based interpretation. Attention matrices can, in principle, reflect how information flows between tokens and capture pairwise dependencies within a head or a single transformer layer. And several studies have used attention patterns to infer cooperative relationships between regulatory features in sequence-based models.[7,18,73] However, attention matrices describe layer- and head-specific weighting during intermediate message passing and are thus highly sensitive to model depth, initialization, and training noise. Their interpretation depends on which layer or head is analyzed, making them difficult to compare across models or biological contexts. In contrast, the final regulator embeddings in ChromBERT are derived from iterative multi-head self-attention, residual aggregation, and nonlinear transformations across all layers, integrating diverse contextual cues into a single stable latent representation. As such, embedding similarity provides a more robust and biologically interpretable measure of regulator relationships.

After fine-tuning ChromBERT to predict regulatory effects in a specific cell type or regulatory dynamics between two cell types, the model assigns each regulator a context-specific embedding within every 1-kb genomic bin. We analyzed these task-adapted, context-specific regulator embeddings to explore (i) key regulators driving regulatory effects in specific cell types, (ii) key regulators driving regulatory dynamics across distinct cellular states, and (iii) context-dependent cooperation among regulators.

Because regulator embedding at individual loci can be noisy, we computed a robust regulator embedding vector by averaging embeddings across a biologically meaningful set of genomic bins $G$, using the following formula:

$$\overline{E}_{G,m}^{\prime(TRN)} = \frac{1}{|G|}\sum_{i \in G} E_{i,m}^{\prime(TRN)}$$

(Equation 7)

where $G$ represent the set of 1-kb genomic bins and $\overline{E}_{G,m}^{\prime(TRN)}$ denotes the average regulator embedding for the regulator $r_m$ across the set of loci $G$ derived from the fine-tuned model.

To identify key regulators underlying cell-type-specific regulatory effects, we focused on regulators exhibiting significant embedding shifts between two groups of genomic bins associated with distinct regulatory outcomes. Specifically, we divided the bins into two groups, $G_1, G_2$, that show contrasting outcomes (e.g., causal vs. non-causal eQTLs), and quantified the embedding shifts for each regulator using 1 – cosine similarity:

$$D_m(G_1, G_2) = 1 - \frac{\overline{E}_{G_1,m}^{\prime(TRN)} \cdot \overline{E}_{G_2,m}^{\prime(TRN)}}{\left\|\overline{E}_{G_1,m}^{\prime(TRN)}\right\|\left\|\overline{E}_{G_2,m}^{\prime(TRN)}\right\|}$$

(Equation 8)

The key principle is straightforward: when a regulator's embedding shifts markedly and consistently between regions with different outcomes, the regulator is likely a key driver of the observed regulatory effect. A greater shift $D_m$ value indicates higher contextual variability and thus greater functional importance in distinguishing the regulatory effects between the two groups. Regulators with the largest $D_m$ values are therefore ranked as key regulators in the given cellular context, enabling ChromBERT to identify context-specific drivers even when direct binding data are sparse. For full task details, see the following Downstream tasks section.

To identify key regulators governing regulatory dynamics between two cell types, ChromBERT applies an analogous procedure. Here, the embedding shifts are calculated between genomic regions that exhibit altered versus unchanged regulatory effects across the two cell types. This approach allows ChromBERT to pinpoint transcription regulators driving cell-type-specific transcriptional dynamics and provides mechanistic insights into the regulatory processes underlying cell-state transitions. Further methodological details are described in the Downstream tasks section.

Context-dependent cooperation between pairs of regulators was quantified directly using the cosine similarity of their embedding vectors.

### Downstream tasks

#### *Hi-C imputation*

We employ pre-trained TRN embeddings to enhance Hi-C contact map imputation, effectively capturing spatial chromatin regulation. In this task, we used pre-trained TRN embeddings to impute high-resolution contact maps from low-resolution Hi-C contact map with a CNN. The input for each region pair consists of three channels: (1) an O/E-normalized low-resolution Hi-C contact map (up-sampled by bilinear interpolation from 5-kb resolution to 1-kb resolution); (2) a pairwise TRN embedding similarity map computed by cosine similarity of pre-trained TRN embeddings for regions pairs; and (3) a pairwise genomic distance map expressed as $(1 + D_g)^{-0.75}$, where $D_g$ represents genomic distance of a region pair, and the exponent $-0.75$ is grounded in prior work modeling chromatin contact decay with genomic distance.[74,75]

These three channels are stacked into an input tensor of shape $(3, N, N)$, where $N = 1{,}000$ (a 1-Mb genomic window at 1-kb resolution). The CNN has three 2D convolution layers: the first expands the three input channels into 64 feature channels using 64 kernels of size $13 \times 13$ (with padding to preserve spatial dimensions); the second further refines these 64 channels using the same kernel size; and the final layer applies a single $13 \times 13$ kernel to collapse the representation into one channel, producing the predicted contact map of shape $(1, N, N)$.

During training, we used a sliding window of 1-Mb (1,000 1-kb bins) moving diagonally in 200-kb steps. To match the target resolution of 1 kb, Hi-C contact maps originally processed at a 5-kb resolution were up-sampled to 1-kb by using two-dimensional linear interpolation. Specifically, we applied bilinear interpolation, which estimates contact values at intermediate 1-kb positions by linearly combining the values of the four nearest bins in both row and column directions.[76] Ground truth was provided by O/E-normalized Micro-C contact maps. Chromosomes were split into training (1, 4, 7, 10, 13, 17, 18), validation (3, 6, 9, 12, 16, 19, 20, X), and test (2, 5, 8, 11, 14, 15, 21, 22) sets. Finally, to reduce bias from regions excluded during ChromBERT pre-training (where regulator binding was assumed all-zero), we adjusted cosine similarity values by replacing them with the average similarity at the same genomic distance.

#### *Cistrome imputation using cell-type-specific prompts (fine-tuning with prompts)*

The current matrix representing cell types and transcription regulators is highly sparse due to the lack of available cistromes for many transcription regulators across diverse cell types. To address this issue, our method integrates cell-type-specific prompts based on chromatin accessibility into the model to impute the matrix by predicting new cistromes, leveraging chromatin accessibility's established role in predicting transcription regulator binding statuses. Alternatively, cell-type-specific prompts based on transcriptomic profiles can be incorporated, as these profiles are also linked to transcription regulator binding statuses and are abundantly available. Furthermore, our approach incorporates a regulator embedding to differentiate among various transcription regulators, facilitating a universal generative model architecture capable of generating cistromes for distinct transcription regulators and performing binary

🔓 **CellPress**

**Cell Genomics**
**Technology**

classification to determine their presence or absence. These prompts improve the model's ability to adapt its predictions to specific cell types.

- DNase-seq prompt: We leveraged DNase-seq data extensively available for various cell types in the Cistrome-Human-6K. For each cell type, ChromBERT's pre-trained embeddings specifically for DNase-seq data were used, denoted as $E_{i,j}^{(DNase)} \in \mathbb{R}^{d_{model}}$, where $i$ represents the $i$-th bin and $j$ indicates the position index of DNase-seq data within the Cistrome-Human-6K dataset for the given cell type.
- RNA-seq prompt: Transcriptome prompts were derived from cell embeddings of scGPT[6] following zero-shot learning on transcriptomic profiles, either from single-cell or bulk cell data, specific to the cell type, denoted as $E^{(scGPT)} \in \mathbb{R}^{d_{scGPT}}$, where $d_{scGPT} = 512$. For single-cell RNA-seq data, cells covering fewer than 200 or more than 7,000 genes were filtered out, and the sum of read counts in each cell were normalized to $1 \times 10^4$ followed by a log1p transformation using Scanpy (v1.9.5)[69]. scGPT then processed these profiles, only considering genes with non-zero expression levels for the computation of attention. For bulk cell RNA-seq data, we utilized the same normalization and transformation procedure and used top 7,000 highly-expressed genes to obtain cell embeddings.

The regulator embedding corresponds to pre-trained regulator embedding $E_{i,m}^{\prime(TRN)}$ for given regulator $r_m$ at the $i$-th 1-kb bin, which is used to distinguish different regulators.

Then for this fine-tuning task, the imputation prompts can be:

$$E_i^{(prompt)} = concat\left(E_{i,m}^{\prime(TRN)}, E_{i,j}^{(DNase)}\right)$$
(Equation 9)

Or

$$E_i^{(prompt)} = concat\left(E_{i,m}^{\prime(TRN)}, E^{\prime(scGPT)}\right)$$
(Equation 10)

based on the choice of cell-type-specific information representation, where $E^{\prime(scGPT)}$ represents transformed $E^{(scGPT)}$ by a linear layer to match the hidden dimensions for concatenation. Remarkably, $E^{\prime(scGPT)}$ is cell representations and is consistent across all bins in a certain cell type.

To fine-tune the model for cistrome prediction across various cell types and transcription regulators using DNase-seq prompts, the model was trained with DNase-seq data from 22 cell types and utilized 104 cistromes corresponding to 29 transcription regulators within these cell types (Table S1). For the model fine-tuned with RNA-seq prompts as cell-type-specific indicators, training was conducted using scGPT cell prompts derived from the RNA-seq dataset, encompassing 20 cell types and 102 cistromes for 29 transcription regulators within these cell types (Table S1). After the training, the fine-tuned model based on DNase-seq prompts or RNA-seq prompts were both applied to perform prediction in bulk cell types for transcription regulators. Additional details of the training settings were summarized in Table S3.

And the fine-tuned model based on RNA-seq prompts was further applied to human PBMC 10× single-cell multi-omics dataset. Paired single-cell RNA-seq and ATAC-seq data were collected from the 10× Genomics multi-omics repository (https://www.10xgenomics.com/datasets/pbmc-from-a-healthy-donor-granulocytes-removed-through-cell-sorting-10-k-1-standard-2-0-0).
Initial data processing and quality control for single-cell RNA-seq data were performed according to protocols in muon-tutorials (https://muon-tutorials.readthedocs.io/en/latest/single-cell-rna-atac/pbmc10k/1-Gene-Expression-Processing.html). We then processed to cluster cells and annotated different cell clusters based on marker gene expression profiles as describe in the same tutorial. Similar data processing and quality control steps were performed for the single-cell ATAC-seq data according to the muon-tutorials (https://muon-tutorials.readthedocs.io/en/latest/single-cell-rna-atac/pbmc10k/2-Chromatin-Accessibility-Processing.html). After these initial quality checks, we further refined the dataset by excluding single-cell ATAC-seq reads detected in fewer than 10 cells and cells with fewer than 1,000 detected genes in single-cell RNA-seq data. With this workflow, we obtained a final 9,493 cells from an initial pool of 11,898 cells. After data processing and cell clustering, we applied fine-tuned ChromBERT with single-cell RNA-seq embeddings as transcriptome prompts to perform prediction in single cells for transcription regulators. Specifically, in this prediction section, we removed all cell clusters related to T cells as we observed that scGPT cell embeddings could not distinguish T cells from other cell clusters well.

### *Fine-mapping of eQTLs with DNA sequence prompt (fine-tuning with prompts)*
To fine-tune ChromBERT for fine-mapping of eQTLs with DNA sequence prompt, we concatenated TRN embeddings with DNA sequence prompt to be fed into an MLP decoder to classify the causal and non-causal eQTLs. For each eQTL, we obtained DNA sequence embeddings with DNABERT-2 for a 1-kb window (from upstream 500 bp to downstream 499 bp of the coordinate of the eQTL). The variant DNA sequence embeddings $E_i^{(variant)}$ were used as the DNA sequence prompt.

For this classification task, we collected eQTLs from the EBI eQTL catalog (https://www.ebi.ac.uk/eqtl) which utilized the SuSiE protocol as described in the Enformer study.[7,77] We systematically analyzed eQTLs across 49 different tissues. For each tissue, variants demonstrating a high likelihood of influence on gene expression, indicated by a posterior inclusion probability (PIP) > 0.9, were categorized as causal variants. Conversely, for each identified causal variant, we sought a corresponding non-causal variant. This non-causal variant was chosen from the set with a PIP <0.01 but an absolute $Z$ score >4 tested for the same gene, wherever available.

In cases where a matched gene variant was not available, we instead selected a random variant from the genome-wide locus with a PIP <0.01 and an absolute $Z$ score >6, to serve as the non-causal counterpart. Following the approach used in Enformer study,[7] we fine-tuned the model in each tissue by conducting 10 iterations of cross validation, maintaining a training/testing ratio of 6:4. Additional details of the training settings were summarized in Table S3. Key differential regulators were then quantified and ranked according to the embedding shift of each regulator between causal and non-causal eQTLs.

### Prediction for genome-wide STARR-seq signals (fine-tuning for cell-type-specific regulatory effects)

For this task, we utilized STARR-seq data from wild type HCT116 cells available in GSE156740.[43] This involved downloading both the STARR-seq signals and a reference set of enhancers, which are based on the human genome build hg19. To align this with ChromBERT, which uses hg38, we converted hg38 1-kb bins to hg19 using the liftOver tool (https://hgdownload.soe.ucsc.edu/admin/exe/linux.x86_64/liftOver). We then associated the human hg38 1-kb bins with the reference STARR-seq enhancers and processed the STARR-seq signals using a $log_2$-transformed fold changes with the pseudo count one. The model was trained specifically on regions that included these reference enhancers and additional open chromatin regions identified by DNase-seq data in HCT116 cells (ENCFF240LRP). This training approach was designed to enhance robustness of our model for variance of STARR-seq signals. Importantly, we excluded all cistromes related to histone modifications or chromatin accessibility to focus specifically on the influence of transcription factors on enhancer activity. Additional details of the training settings were summarized in Table S3.

### Prediction for perturbation effect on STARR-seq signals (fine-tuning for cell-type-specific regulator effects)

For predicting the perturbation effects on STARR-seq signals, we adapted ChromBERT to predict changes in control and 3-indoleacetic acid (IAA)-treated HCT116 cells. IAA treatment is known to rapidly deplete tagged transcription factors.[43] We used the original reference cistromes for control cell predictions and applied perturbations for binding status of depleted factors in treated cells (see the perturbation of cistromes section above). Training for this perturbation model included STARR-seq signals in both control (-IAA) and treated (+IAA) conditions for three transcription factors: BRD2, CDK7, and EP300/CREBBP. Unlike the model for wild type signals, this perturbation model was trained using reference STARR-seq enhancers that show significant variance in response to treatment. The top 30 regulators showing the highest embedding similarity with EP300 across all test enhancers prior to perturbation were identified as potential EP300 cofactors. We then quantified the embedding shifts of cofactors between wild-type and EP300-perturbed conditions at enhancers that showed decreased STARR-seq activity upon EP300 perturbation, enabling detection of cofactors whose function is modulated by EP300.

### Classification of EZH2's classical and non-classical sites (fine-tuning for cell-type-specific regulatory effects)

For this classification task, we used EZH2 (GSE29611[2]) and H3K27me3 (GSE61176) ChIP-seq data from human embryonic stem cells. We labeled bins overlapping with both EZH2 and H3K27me3 as positive and those only overlapping with EZH2 as negative to create a clear binary classification dataset. As H3K27me3 was included in the pre-trained dataset and was proposed to have dominant effect on the model output, we omitted cistromes associated with H3K27me3 from the input reference cistromes. Additional details of the training settings were summarized in Table S3. Cofactors of EZH2 at classical or non-classical sites were then quantified and ranked according to the cosine similarity between each regulator's embedding and that of EZH2 within the corresponding site set.

### Prediction for genome-wide changes in transcriptome or chromatin accessibility (fine-tuning for regulatory dynamics between cell types)

In this task, we fine-tuned ChromBERT to predict genome-wide changes in transcriptome or chromatin accessibility in cellular heterogeneity and during transdifferentiation. Specifically, we quantified genome-wide changes in transcriptome with fold changes of gene expression levels, and quantified genome-wide changes in chromatin accessibility through fold changes of DNase-seq or ATAC-seq signals. Additional details of the training settings were summarized in Table S3.

After data processing and cell clustering for human PBMC 10× single-cell multi-omics dataset mentioned in the above sections in STAR Methods. We specifically assigned cell clusters into five major cell types for this prediction section, including T cells, B cells, monocytes, dendritic cells (DCs) and natural killer (NK) cells. This assignment helps us to collected well-known key regulators for each cell type from the public studies. For each cell type, single-cell RNA-seq data were aggregated into pseudo-bulk counts and normalized using a factor of $1 \times 10^6$. Single-cell ATAC-seq data were also compiled into pseudo-bulk reads and analyzed for peak calling using pycisTopic (v1.0.3),[64] and these reads were normalized to a count of $1 \times 10^6$. Then we fine-tuned ChromBERT to predict changes in transcriptome or chromatin accessibility between cell types in a pairwise manner. For transcriptome profiles, we calculated changes using differences of log1p-transformed pseudo-bulk expression level across all genes. For chromatin accessibility, changes were calculated by $log_2$-transformed fold changes in pseudo-bulk reads density at merged peaks and background regions (spanning 10 kb upstream and downstream around all TSSs), with a pseudo count one in fold change calculation.

For transcriptome changes in transdifferentiation, we collected transcriptome profiles derived from CAGE-seq in seven cell types available in the FANTOM5 database,[78] including fibroblasts, induced pluripotent stem cells (iPSCs), myoblasts, hepatocytes, macrophages, cardiac myocytes, and B cells. We focused on five transdifferentiation processes: fibroblast to iPSCs, fibroblast to myoblast, fibroblast to hepatocyte, fibroblast to cardiac myocyte, and B cells to macrophage. Bulk-cell transcriptome changes across different cell states were quantified with differences of log1p-transformed gene expression level (TPM) across all genes.

For chromatin accessibility changes in transdifferentiation, we utilized DNase-seq in fibroblast (ENCFF184KAM), iPSCs (ENCFF540VPT), myoblast (ENCFF647RNC), cardiac myocyte (ENCFF389SOW), B cells (ENCFF650BNV) and macrophage (ENCFF580ICE), and ATAC-seq in fibroblast and hepatocyte (GSE179011,[79] GSE185358[80]). Peak calling and reads density

normalization were performed in the approaches similar to those used in our single-cell ATAC-seq pseudo-bulk processing. And bulk-cell chromatin accessibility changes were also quantified by $\log_2$-transformed fold changes in reads density at merged peaks and background regions (spanning 10 kb upstream and downstream around all TSSs), with a pseudo count one in fold change calculation.

We performed interpretation analysis to identify key regulators in cellular heterogeneity and during transdifferentiation. In the realm of cellular heterogeneity, we categorized genes and loci based on elevated or stable expression and chromatin accessibility. Specifically, genes exhibiting a log1p-transformed expression difference greater than 1 were classified as having increased expression, while those with a difference ranging from $-0.5$ to 0.5 were considered stable. In chromatin accessibility analysis using ATAC-seq, regions with a $\log_2$-transformed increase greater than 2 were marked as having increased chromatin accessibility. After excluding regions that had no coverage, we selected the top 40,000 regions showing the minimal absolute fold changes as having unchanged chromatin accessibility. The method for defining genes and regions with increased or unchanged expression or accessibility during transdifferentiation was analogous to that used for analyzing cellular heterogeneity.

Each analysis between two single-cell pseudo-bulk cell types in cellular heterogeneity or two cell states in a transdifferentiation process involved ranking transcription factors based on their embedding shift in two set of loci for transcriptome or chromatin accessibility dynamics, respectively. Epigenetic modifications were not considered in this analysis. We averaged ranking scores from these two modalities to compute a composite ranking score for 991 transcription factors, identifying those most influential in driving cellular state transition.

In our analysis of cell state transition in each transdifferentiation process, the top 25 transcription factors were recognized as key regulators of this transdifferentiation process. For cellular heterogeneity, we computed an average ranking score for each regulator by averaging ranking scores for the given cell type relative to each other (defined as cell-type-specific ranking scores). This detailed evaluation helps us understand the importance of these key regulators in differentiating the given cell type from all other cell types and identify the top 25 key regulators for each cell type, 84 key regulators in total. And cell-type-specific ranking scores for all 84 key regulators help us to further assign the cell-type-specificity of identified key regulators, which were generally classified into four categories, B cells-specific, T cells/NK cells-specific, monocytes-specific and DCs-specific, as illustrated in Figure 4B.

### Benchmarks
#### *Benchmark for cistrome imputation*
We assessed the performance of ChromBERT on the cistrome imputation task by comparing it with both Avocado and a baseline approach based on candidate peaks and chromatin accessibility. For the Avocado comparison, we used imputed tracks from ENCODE (ENCSR481OSA) and restricted our analysis to cistromes withheld from the fine-tuning of ChromBERT and the training of Avocado. For benchmarking, imputed signals derived from BigWig files were aggregated into 1-kb bins.

To build the baseline, we combined a regulator's candidate peaks with chromatin accessibility information. Candidate peaks were defined in two ways. When at least five cistromes for the regulator (excluding the test cistrome) were available in the Cistrome-Human-6K dataset, we collected all corresponding peaks and merged them into 1-kb bins to form the candidate peak set. When fewer than five cistromes were available, candidate peaks were instead defined as 1-kb bins containing the regulator's sequence motif. This procedure generates a binary vector $B \in \mathbb{R}^N$ (where $N = 2,137,894$ denotes all genomic 1-kb bins), with $B_i = 1$ indicating that bin $i$ is a candidate peak for the regulator. To obtain cell-type-specific binding probability ranks, we then weighted these candidate peaks by multiplying $B$ with DNase-seq signal from the corresponding cell type in an element-wise approach.

We evaluated ChromBERT's performance to predict transcription regulator binding sites that are specific to a particular cell type, even when chromatin accessibility is similar across cell types. To define these sites, we performed pairwise comparisons between paired cistromes from different cell types but associated with the same transcription regulator. A binding site was classified as cell-type-specific if it was present in one cell type but absent in the other, represented as 1-kb bins. For a site to be considered peak-present, it had to meet two criteria: (1) the signal intensity had to exceed 3; (2) the signal had to be at least 3-fold higher than in the peak-absent cell type. To control for potential differences in chromatin accessibility, we applied additional constraints: (1) the absolute $\log_2$-transformed fold change in DNase-seq signal between the two cell types had to be less than 0.5; (2) the DNase-seq signal in both cell types had to exceed 3, ensuring that both sites were located in accessible chromatin. ChIP-seq and DNase-seq signals were normalized to the genomic background based on read densities. To ensure statistical robustness, we included only cistrome pairs with at least 50 cell-type-specific binding events, while also requiring that the ratio of peak-present to peak-absent bins remained balanced, with a maximum difference of less than 5-fold. ChromBERT's performance to distinguish cell-type-specific binding events was assessed by comparing its predicted probability differences between paired cell types against the ground truth—defined by the actual presence or absence of transcription regulator binding based on ChIP-seq data.

#### *Benchmark with enformer and DNABERT-2*
In our study, we also compared the performance of ChromBERT with two advanced DNA sequence-based models, Enformer[7] and DNABERT-2,[8] in a subset of our downstream tasks which can also be accomplished by the two models based on DNA sequences (genome build hg38). These models, which generate contextual representations for input DNA sequences, were fine-tuned using the established method in the original studies with slight modification to suit specific requirements of the tasks. For DNABERT-2, we implemented a fine-tuning protocol available from the study (https://github.com/MAGICS-LAB/DNABERT_2/blob/main/finetune/train.py). In the case of Enformer, we used a random forest approach, either as a classifier or a regressor, to perform predictions on

downsteam tasks based on the embeddings generated from DNA sequences. The random forest classifier and regressor were implemented using the package scikit-learn (v1.3.0),[81] with the "max_features" parameter set to "log2". The number of frozen transformer layers, loss functions and training hyperparameters for DNABERT-2 and Enformer were summarized in Table S3.

Both models were trained using the same dataset as ChromBERT. DNABERT-2 processed the entire DNA sequences in 1-kb bins, similar to ChromBERT. If ChromBERT was set to consider multiple adjacent bins (e.g., $n = 9$ bins for changes in the cellular transcriptome), DNABERT-2 followed the same approach and then applied a max pooling strategy to the embeddings. Enformer, designed to handle long-range DNA sequences, required inputs of 196,608 bp. For this purpose, we used DNA sequences centered around the 1-kb bins used in ChromBERT. These inputs were then transformed into 896 sequence vectors by Enformer, corresponding to 114,689 bp in 128-bp resolution. To match the lengths of input regions from different models, we selected the central 8 out of 896 sequence vectors from Enformer's output, corresponding to 1,024 bp, as Enformer's embeddings for the given 1-kb bin. For these 8 sequence vectors, we performed a mean pooling strategy to generate an averaged sequence vector for the following random forest.

For tasks where ChromBERT used multiple adjacent bins (e.g., $n = 9$ bins for changes in the cellular transcriptome), we selected the central 70 out of 896 sequence vectors from Enformer's output, corresponding to 8,960 bp, and applied a similar mean pooling strategy. The training configurations for each model were summarized in Table S3, and the runtime comparisons between ChromBERT and DNABERT-2 were provided in Table S4.

### Gene ontology enrichment analysis

We conducted Gene Ontology (GO) enrichment analysis using the DAVID online tool (https://david.ncifcrf.gov/tools.jsp).[82] This analysis assessed the enrichment of human gene ontology biological process (BP) terms.

### Motif scan

Motif scans were performed using FIMO (v5.0.5)[69] against the JASPAR core 2024 vertebrates database[70] with the following parameters "–max-stored-scores 1000000". Motifs with $p$-value $\leq 1 \times 10^{-5}$ were used for the following analysis.

### QUANTIFICATION AND STATISTICAL ANALYSIS

Statistical details are indicated in the corresponding figure legends. Statistical significance was defined and represented as $p$-value <0.05 (*), $p$-value <0.01 (**), $p$-value $<1 \times 10^{-3}$ (***) and $p$-value $<1 \times 10^{-4}$ (****) from a two-sided Mann-Whitney $U$-test unless otherwise specified in the figure legends.

