## [Document S2. Transparent peer review records for Yu et al. · Cell Genomics]

ChromBERT: A foundation model for learning interpretable representations for context-specific transcriptional regulatory networks

Author list: Zhaowei Yu, Dongxu Yang, Qianqian Chen, Yuxuan Zhang, Zhanhao Li, Yucheng Wang, Chenfei Wang, Yong Zhang

Summary

Initial submission: Received : 23rd July 2025

Scientific editor: Judith Nicholson

First round of review: Number of reviewers: 2
Revision invited : September 10th 2025
Revision received : November 6th 2025

Second round of review: Number of reviewers: 2
Accepted : 11th December 2025

Data freely available: N/A

Code freely available: Yes

This transparent peer review record is not systematically proofread, type-set, or edited. Special characters, formatting, and equations may fail to render properly. Standard procedural text within the editor's letters has been deleted for the sake of brevity, but all official correspondence specific to the manuscript has been preserved.

Referees' reports, first round of review

Reviewer 1:

Comments

This study introduces ChromBERT, a transformer-based foundation model pre-trained on large-scale ChIP-seq data to learn transcription regulator interactions across the genome. Through prompt-enhanced fine-tuning, the model achieves strong performance on multiple downstream tasks, including cistrome imputation, eQTL fine-mapping, enhancer activity prediction, and cell state transition analysis. A key strength is its interpretability: fine-tuned embeddings can be compared across conditions to identify potential key regulators without additional ChIP-seq data.

However, there are several points that require clarification or further discussion:

Major Comments

1) The authors adopt a 768-dimensional hidden vector and a 1 kb window size as the basic input representation for ChromBERT. However, the rationale for these design choices is not clearly explained.

What is the theoretical or empirical justification for using a 768-dimensional embedding? Was this value chosen based on hyperparameter optimization, or simply borrowed from the BERT architecture in NLP?

The choice of a 1 kb genomic window also requires further clarification:

- Why not use a larger window size (e.g., 2 kb, 4 kb, or even longer) to capture more distal regulatory signals?
- Is a 1 kb window sufficient to encompass typical enhancer-promoter interactions, especially those involving long-range chromatin contacts?

Have the authors evaluated the impact of different window sizes on model performance? It would be valuable to include comparative experiments or additional discussion in the supplementary materials to justify this choice.

2) According to the manuscript, the 768-dimensional embedding vector encodes both the binding status of the transcription factor and the contextual influences of other regulators in the same region via the self-attention mechanism. This raises several questions:

- Can the authors analyze or visualize these embedding vectors (e.g., through dimensionality reduction) to explore whether they capture distinct patterns related to proximal versus distal regulatory signals?
- Could differences in embedding shifts across genomic regions be leveraged to identify signals associated with long-range enhancer activity?
- If the embeddings indeed encode distal regulatory information, would the authors consider providing an analysis linking embedding dimensions to known cis-regulatory distances or chromatin interaction data (e.g., Hi-C or ChIA-PET) to further support how the model captures long-range regulatory mechanisms?

3) The authors show that ChromBERT's pre-trained TRN embeddings capture regulatory architecture, local chromatin structure, and even 3D genome organization. They also demonstrate that regulator embeddings reflect functional cooperation among transcription factors. However, the manuscript does not clearly address how much of this information is directly learned from ChIP-seq binding patterns versus indirectly inferred through co-occurrence biases in the training data. While the pre-trained embeddings show promising associations with regulatory architecture and TF cooperation, relying solely on embedding similarity might conflate biological signal with statistical co-occurrence.

To strengthen the claims, it would be valuable for the authors to:

- Perform control experiments using shuffled or cell-type-permuted binding data to confirm that the observed clustering and 3D contact associations are not driven by global frequency biases of transcription factors.
- Authors can calculate the correlation between embedding similarity and simple metrics such as co-occurrence frequency, Jaccard similarity, and mutual information to confirm whether embedding provides additional information gain.

Such analyses confirm that embeddings indeed capture context-dependent functional interactions beyond simple co-occurrence patterns, and which would better support the conclusion that ChromBERT learns context-dependent regulatory architecture beyond simple binding co-occurrence patterns.

4) The manuscript leverages DNase-seq and scGPT-derived transcriptome-based prompts to align cell-type-specific chromatin accessibility and transcriptional states with ChromBERT embeddings, enabling cistrome imputation in unseen cell types. However, it remains unclear why aligning these two modalities should provide sufficient information to predict cell-type-specific binding sites of transcription factors, especially in cases where chromatin accessibility is similar across cell types.

To strengthen this claim, the authors are encouraged to:

- Provide references to prior studies that demonstrate a strong predictive link between DNase-seq or transcriptomic profiles and transcription factor binding specificity.
- Perform ablation studies or comparative experiments testing different prompt sources (e.g., DNase-seq only, scGPT only, or combined prompts) to quantify their respective contributions to cistrome imputation.
- Validate whether the improved performance on cell-type-specific binding sites persists when controlling for chromatin accessibility, by comparing predictions with and without incorporating prompt information.

5) The three downstream tasks (eQTL fine-mapping, enhancer activity prediction, and EZH2 functional mode classification) share a common framework: ChromBERT is first fine-tuned on task-specific objectives to obtain context-adapted TRN embeddings, and then regulator embeddings are interpreted via embedding shift analysis to identify key regulators. While this unified approach is a strength of the study, several aspects require clarification and further validation:

- Clarify methodological consistency and differences. Although the overall

pipeline is similar, the manuscript does not explicitly explain whether the same fine-tuning strategy (e.g., loss functions, training epochs, prompt usage) is applied across all tasks or if adjustments are made for each task. A brief comparison of how fine-tuning is tailored to different objectives (classification vs. regression) would improve reproducibility.

- Explain how embedding shift is quantified and validated. The interpretation step is central to all three tasks, yet the manuscript lacks details on how embedding shift is computed (e.g., distance metrics, normalization) and whether results are robust to alternative similarity measures. It would strengthen the claims to include validation experiments, such as comparing embedding shift-derived regulators with known functional regulators from perturbation datasets.

- Discuss the generalizability of the interpretation framework. While the three tasks demonstrate the versatility of the approach, the authors should discuss potential limitations when applying this pipeline to other tasks or regulators with sparse training data. For example, does the fine-tuning process risk overfitting when only a small set of labeled examples is available?

6) The authors present ChromBERT as a powerful framework for fine-tuning pre-trained TRN embeddings to predict transcriptional regulatory outcomes and identify key regulators through embedding shift analysis.

ChromBERT inherently learns contextual relationships among transcription factors via self-attention mechanisms, with each TRN embedding encoding a region-specific representation of regulator interactions. The regulator embeddings can be regarded as node-specific contextual features, while self-attention weights or embedding similarities may implicitly reflect co-regulatory relationships between transcription factors. These properties suggest that ChromBERT could be extended beyond key regulator ranking to systematically infer transcriptional regulatory networks (TRNs) at cell-type- or state-specific resolution.

While the results convincingly demonstrate its predictive and interpretive capabilities, the study would benefit from a deeper discussion of ChromBERT's potential for explicit regulatory network reconstruction, which could significantly broaden its impact.

Given that TRN embeddings encode region-specific contextual interactions among transcription factors, several methodological directions could be explored:

Utilizing self-attention weights and embedding similarities - The transformer architecture inherently models dependencies among regulators, and the resulting attention weights or embedding similarity matrices could serve as a basis for inferring regulator-regulator connections. Aggregating these relationships across genomic regions might allow the construction of cell-type-specific regulatory interaction networks.

Combining embeddings with external evidence - To ensure biological relevance and directionality, predicted edges could be filtered or weighted using additional information such as motif presence, ChIP-seq binding profiles, or perturbation-based functional screens.

Minor Comments

1) The schematic in Figure 2 is conceptually unclear, particularly in differentiating regulator embeddings from TRN embeddings and showing how these two representations interact during prompt-enhanced fine-tuning. The current layout makes it difficult to follow the flow of information and understand how prompts are integrated into the model to enable regulator-specific cistrome predictions.

To improve clarity, the authors should:

Clearly define and visually distinguish regulator embeddings versus TRN embeddings, explaining their respective roles.

Provide a more structured diagram that explicitly illustrates how prompts are combined with these embeddings during fine-tuning and imputation.

Ensure that the logical flow of the figure aligns with the textual description, making it easier to understand the relationships among prompts, embeddings, and final predictions.

2) The manuscript demonstrates that DNase-seq-based chromatin accessibility and scGPT-derived transcriptome prompts improve ChromBERT's ability to impute cell-type-specific cistromes. However, the relative contributions of chromatin accessibility information and gene expression information to predicting cell-type-specific transcription factor binding remain unclear.

The authors should discuss:

To what extent each modality (DNase-seq vs. transcriptome-based prompts) contributes to model performance, and whether one source provides more cell-

type-specific information than the other.

The biological plausibility of these contributions, supported by references to prior studies showing correlations between accessibility, expression, and TF binding.

Potential differences in how these modalities might complement each other, particularly for factors whose binding is not strictly determined by accessibility alone.

3) Figure 3A provides an overview of how fine-tuned TRN embeddings are used for interpretation via embedding shift, but the schematic is highly abstract and does not clearly illustrate the workflow for each downstream task (eQTL fine-mapping, enhancer activity prediction, EZH2 functional mode classification).

To improve clarity, the authors could extend Figure 3A or provide supplementary subpanels that explicitly depict the task-specific inputs and labels used for fine-tuning in each task (e.g., causal/non-causal variants for eQTL, enhancer activity values for STARR-seq).

Such a figure would better illustrate the versatility of the approach and help readers understand the specific workflows for each application.

Reviewer 2:

The work describes ChromBERT, a deep learning model of chromatin state trained in a self-supervised fashion to predict masked tokens within 1 kb regions in the human genome. The resulting model is evaluated on its ability to serve as a platform for training accurate models for other regulatory genomics applications. The approach is novel and is complementary to the more traditional sequence-based approach.

Strengths:

* Well thought out deep learning approach that is unique in its modeling approach. The idea of adding additional external data they refer to as "prompts" is also novel, and allows the model to incorporate complementary information e.g. from DNA LLMs.

* Comprehensive evaluation on several gene regulation tasks demonstrate the

advantage of fine-tuning ChromBERT over fine-tuning sequence-based models (Enformer and DNABERT2).

* The manuscript is well-written and relatively easy to understand (except when it isn't - see comments below).

Weaknesses:

* Some aspects of the model need to be explained better. For example, the description of the decoder architecture during pre-training; Hi C imputation also needs to be explained better. In this case, it's not clear how the CNN and genomic distance map were used (and why a power of -0.75, which seems rather arbitrary).

* For the EZH2 task, "ChromBERT was fine-tuned by omitting cistromes associated with H3K27me3 from input reference cistromes to diminish its dominant role in the task." That may not be sufficient, as the model will have been trained on those cistromes, and so is given an unfair advantage over DNABERT2 and Enformer (see also the comment on training, validation and test sets).

* Functional collaboration between regulators is assessed by the similarity of their embeddings. While this heuristic is intuitive, there is nothing in the network that would lead this to hold in general. A common way to establish associations between regulators uses the attention matrix. This has been demonstrated to work well in the context of sequence-based deep learning, and is likely to work here as well. See e.g.:

Ullah, Fahad, and Asa Ben-Hur. "A self-attention model for inferring cooperativity between regulatory features." *Nucleic acids research* 49.13 (2021): e77-e77.

Zhang, Zhenhao, et al. "A generalizable framework to comprehensively predict epigenome, chromatin organization, and transcriptome." *Nucleic Acids Research* 51.12 (2023): 5931-5947.

* Sequence based models generate embeddings that are useful for tasks across species. It is not clear if the embeddings generated by ChromBERT (trained in human) are useful in other species, e.g. mouse.

Additional Comments:

* It is interesting that ChromBERT outperforms Enformer, which is trained on very long sequences, whereas ChromBERT is trained on short 1 kb regions. Please comment on that!

* A comparison with the "Sei" model for example, might be instructive (but not necessary) since it is trained on much shorter sequences, and its training data also consists of a similar mix of chromatin state profiles.

* Please add running times of fine-tuning ChromBERT in comparison with time for fine-tuning other models.

* The decay in embedding similarity with distance shown in figure S2B seems quite minor. Please provide details in the manuscript on how that was quantified. Is it the average cosine similarity over all pairs of embedding vectors with a distance falling within a given bin? If that is the case, it would seem all the embeddings are quite similar. Vectors in high dimensional spaces tend to exhibit very low similarity in general, so the level of overall similarity shown in the figure is very surprising.

* There are several sequence-based deep learning models that are designed to generalize to unseen cell types, and are worth mentioning:

Javed, Nauman, et al. "A multi-modal transformer for cell type-agnostic regulatory predictions." *Cell Genomics* 5.2 (2025).

The name of the model (EpiBERT) is VERY similar to ChromBERT, and reflects some similarity in the approach.

Zhang, Zhenhao, et al. "A generalizable framework to comprehensively predict epigenome, chromatin organization, and transcriptome." *Nucleic Acids Research* 51.12 (2023): 5931-5947.

* There is no clear description of the choice of training, validation, and test sets in most cases. It is important that all tasks use consistent choices in this regard to avoid information leakage.

* Please clarify the reasoning for using positional embeddings on top of categorical embeddings. For DNA sequences it helps establish an order relationship; in this case the notion of order is not present, so the rationale is not clear. Is this simply to establish some identifier for each element?

* The github repository contains a mouse version of the model. I did not see it mentioned in the manuscript. Please provides some details on that model!

Typos and grammar:

* PyTorch Lightning (v2.0.4) was implemented... I think you meant "used"

* "In our study, we also compared the performance of ChromBERT with two advanced deep language models, Enformer and DNABERT-2" - Language models are trained in a self-supervised fashion to predict masked tokens. The Enformer is trained to predict gene expression, so is not a language model.

Authors' response to the first round of review

Reviewers' Comments:*Reviewer #1: Comments*

This study introduces ChromBERT, a transformer-based foundation model pre-trained on large-scale ChIP-seq data to learn transcription regulator interactions across the genome. Through prompt-enhanced fine-tuning, the model achieves strong performance on multiple downstream tasks, including cistrome imputation, eQTL fine-mapping, enhancer activity prediction, and cell state transition analysis. A key strength is its interpretability: fine-tuned embeddings can be compared across conditions to identify potential key regulators without additional ChIP-seq data.

However, there are several points that require clarification or further discussion.

Major Comments

1) The authors adopt a 768-dimensional hidden vector and a 1 kb window size as the basic input representation for ChromBERT. However, the rationale for these design choices is not clearly explained.

What is the theoretical or empirical justification for using a 768-dimensional embedding? Was this value chosen based on hyperparameter optimization, or simply borrowed from the BERT architecture in NLP?

Response:

We thank you for raising this point. We adopted a 768-dimensional hidden embedding size for ChromBERT based on the widely used BERT-base architecture in natural language processing (NLP), which provides a well-balanced trade-off between computational efficiency and representational capacity. While this dimensionality was not independently re-optimized for genomic inputs, it provides sufficient capacity to capture the complex combinatorial binding patterns of transcription regulators without incurring excessive computational cost. We acknowledge that different embedding dimensions may affect model performance, and future studies could explore this further through systematic tuning. We have updated the Methods section to clarify the rationale for this design decision and added a note in the Discussion to acknowledge this as a potential improvement point.

The choice of a 1 kb genomic window also requires further clarification

- Why not use a larger window size (e.g., 2 kb, 4 kb, or even longer) to capture more distal regulatory signals?

- Is a 1 kb window sufficient to encompass typical enhancer-promoter interactions, especially those involving long-range chromatin contacts?

Have the authors evaluated the impact of different window sizes on model performance? It would be valuable to include comparative experiments or additional discussion in the supplementary materials to justify this choice.

Response:

We thank you for raising this important point regarding the choice of genomic window size. Our decision to adopt a 1-kb window was motivated by both biological rationale and empirical evidence, as outlined below.

From a biological perspective, ChromBERT is designed to model the context-dependent co-association of transcription regulators within individual *cis* regulatory elements (CREs), rather than to capture long-range chromatin interactions. We chose a 1-kb window size because most transcription regulator co-binding events occur within a few hundred base pair. A 1-kb bin has been widely adopted in genome-wide studies of transcriptional regulation, as it provides an appropriate resolution for capturing local cooperative or competitive interactions among regulators. We have updated our Methods section to clarify this rationale.

We agree that larger windows (e.g., 2 kb or 4 kb) may capture certain short-range enhancer-promoter interactions; however, they also merge signals from neighboring CREs, which may introduce false-positive associations in context-dependent modeling. To systematically evaluate the impact of window size as you suggested, we pre-trained ChromBERT models using 200 bp, 2 kb, and 4 kb windows, and assessed their performance across three downstream tasks (Figures R1A-C). In the STARR-seq and EZH2 prediction tasks, we observed a modest performance decrease as window size increased. We hypothesized that this decline might partly result from differences in the number of training samples (since finer windows yield more training regions). To control for this, we repeated the experiments with down-sampled training datasets to ensure equal sample sizes across models. Under this condition, models trained with 200-bp and 1-kb windows performed comparably and consistently outperformed those pre-trained with 2-kb and 4-kb windows (Figures R1D, E).

While the 200-bp model showed slightly better accuracy, likely due to the increased number of samples, the computational cost was substantially higher (as 200-bp windows produce roughly twice as many training samples). Therefore, we retained 1 kb as a balanced choice between biological resolution and computational efficiency in this study. To promote reproducibility and future research, we have released pre-trained model checkpoints with various window sizes in our GitHub repository. This enables users to select the resolution most appropriate for their specific applications.

Figure R1. Performance of ChromBERT models pre-trained with different genomic window sizes across three downstream tasks. Panels (A-E) display results from ChromBERT models pre-trained on various window sizes, including 200 bp, 1 kb, 2 kb and 4 kb. The notation (n=x/y) indicates the number of positive and negative samples in the training dataset of classification task, while (n=x) indicates the total number of samples in the training dataset of regression task. (A) Bar plots show the performance of these pre-trained ChromBERT models with different window sizes in classifying EZH2's classical and non-classical sites. For this task, ChromBERT was fine-tuned by omitting cistromes associated with H3K27me3 to reduce its dominant influence. (B) Bar plots illustrate the performance of these ChromBERT models with different window sizes in modeling STARR-seq signals in wild-type HCT116 cells. (C) Box plots depict the performance of these pre-trained ChromBERT models with different window sizes in classifying causal versus non-causal expression quantitative trait loci (eQTLs) across 49 human tissues. The statistical significance was assessed between ChromBERT pre-trained with 200 bp window size versus other window sizes by a two-sided Mann-Whitney *U*-test, and **** represents p -value $< 1 \times 10^{-4}$, ** represents p -value < 0.01 , n.s. represents non-significant. The notation (n=x/y) represents the average number of positive and negative samples in the training dataset across 49 human tissues. (D) Bar plots show the performance in classifying EZH2's classical and non-classical sites after down-sampling. Samples were down-sampled to approximately equalize the number by randomly selecting one sample (200 bp, 1 kb, 2 kb) per overlapping 4-kb genomic region. (E) Bar plots illustrate the performance in modeling STARR-seq signals in wild-type HCT116 cells after down-sampling, using the same approach as in panel D.

2) According to the manuscript, the 768-dimensional embedding vector encodes both the binding status of the transcription factor and the contextual influences of other regulators in the same region via the self-attention mechanism. This raises several questions:

- Can the authors analyze or visualize these embedding vectors (e.g., through dimensionality reduction) to explore whether they capture distinct patterns related to proximal versus distal regulatory signals?

Response:

We thank you for this constructive suggestion. In ChromBERT, each genomic region is represented by a transcriptional regulatory network (TRN) embedding matrix with shape (number of regulators x hidden dimension). Each row in this matrix corresponds to a specific regulator and encodes its context-dependent role, integrating information from all other regulators in the same region via the self-attention mechanism. These embeddings therefore capture not only the binding status of individual regulators but also their cooperative and competitive interactions. For region-level analyses shown in Figure S2 (e.g., visualization of local regulatory architecture or comparing regulatory similarity between regions), we generated a single TRN embedding vector for each region by averaging across all regulators. We revised the Methods and Figure Legends sections to clarify this aggregation strategy.

To evaluate whether these region-level embeddings capture distinct regulatory patterns between promoter-proximal and distal regions, we followed your suggestion and performed dimensionality reduction using UMAP. Specifically, we analyzed the averaged TRN embedding vectors from two categories of accessible regions on chromosome 1: (i) promoter-proximal regions (± 500 bp around accessible TSSs in hESCs) and (ii) distal regions (accessible regions in hESCs located more than 10 kb from the nearest TSS). As shown in Figure R2, proximal and distal regions formed partially distinct clusters, suggesting that ChromBERT embeddings capture meaningful differences in regulatory architecture between these categories. We have incorporated this analysis to the revised manuscript as updated Figure S2B.

Figure R2. UMAP visualization of averaged TRN embedding vectors from promoter-proximal and distal regions in hESCs. Each point represents a genomic region on chromosome 1, with embeddings obtained by averaging TRN embeddings across regulators within that region. Promoter-proximal regions are defined as ± 500 bp around accessible TSSs, while distal regions are defined as accessible regions located more than 10 kb away from the nearest TSS. Chromatin accessibility was determined using ATAC-seq data from hESCs (GEO accession: GSM2386582¹).

- Could differences in embedding shifts across genomic regions be leveraged to identify signals associated with long-range enhancer activity?

- If the embeddings indeed encode distal regulatory information, would the authors consider providing an analysis linking embedding dimensions to known cis-regulatory distances or chromatin interaction data (e.g., Hi-C or ChIA-PET) to further support how the model captures long-range regulatory mechanisms?

Response:

We thank you for these insightful suggestions. We performed additional analyses, and our results support the association between high TRN embedding similarities and long-range enhancer activities and regulatory interactions.

While distal regions generally exhibited greater embedding shifts relative to promoter-proximal regions (Figure R2), we hypothesized that functionally connected distal enhancers and promoters (*i.e.* those that form long-range chromatin contacts) should share similar regulatory architectures. Consequently, these pairs would be expected to display lower embedding shift and higher TRN embedding similarity. To test this, we first analyzed experimentally validated long-range enhancers of genes in hESCs. We found that these enhancers exhibited significantly higher TRN embedding similarity to their target promoters compared to other distal accessible regions (Figure R3A). This observation suggests that ChromBERT embeddings can distinguish long-range enhancers from unrelated distal accessible regions based on their embedding similarity to target promoters.

We then extended this analysis genome-wide by comparing promoter-distal region pairs with and without known chromatin contacts. As shown in Figure R3B, distal regions engaged in promoter-enhancer interactions (as identified by Micro-C data in hESCs) exhibited significantly higher TRN embedding similarity to their corresponding promoters. This finding supports the notion that ChromBERT captures context-dependent features relevant to 3D chromatin organization and long-range transcriptional regulation.

Together, these results demonstrate that TRN embedding similarity reflects biologically meaningful long-range regulatory interactions. We incorporated these analyses into the revised manuscript as Figures S2F and S2G.

Figure R3. Association between distal-promoter interactions and TRN embedding similarity. (A) UCSC Genome Browser views showing representative enhancer-promoter contacts for *MOB3C* (chr1:46,539,804-46,780,591; left) and

RNVU1-15 (chr1:144,400,267-144,583,599; right), along with their embedding similarities. Tracks from top to bottom: ATAC-seq signal in hESCs (GSM2386582¹; green), cosine similarity between the ChromBERT embedding of the 1-kb TSS bin (boxed) and nearby accessible distal regions, and chromatin loops detected by Pol II ChIA-PET (ENCSR782EKZ; red arcs). Only chromatin loops linking accessible distal regions to the target promoters are shown. (B) Line plots showing embedding similarity between distal-promoter region pairs. Dark green and light purple lines indicate distal-promoter pairs with and without chromatin contacts, respectively, as determined by Micro-C data in hESCs.

3) The authors show that ChromBERT's pre-trained TRN embeddings capture regulatory architecture, local chromatin structure, and even 3D genome organization. They also demonstrate that regulator embeddings reflect functional cooperation among transcription factors. However, the manuscript does not clearly address how much of this information is directly learned from ChIP-seq binding patterns versus indirectly inferred through co-occurrence biases in the training data. While the pre-trained embeddings show promising associations with regulatory architecture and TF cooperation, relying solely on embedding similarity might conflate biological signal with statistical co-occurrence.

To strengthen the claims, it would be valuable for the authors to:

- Perform control experiments using shuffled or cell-type-permuted binding data to confirm that the observed clustering and 3D contact associations are not driven by global frequency biases of transcription factors.*
- Authors can calculate the correlation between embedding similarity and simple metrics such as co-occurrence frequency, Jaccard similarity, and mutual information to confirm whether embedding provides additional information gain.*

Such analyses confirm that embeddings indeed capture context-dependent functional interactions beyond simple co-occurrence patterns, and which would better support the conclusion that ChromBERT learns context-dependent regulatory architecture beyond simple binding co-occurrence patterns.

Response:

We thank you for these insightful comments and valuable suggestions. We fully agree that it is important to distinguish whether ChromBERT's learned representations reflect true, context-dependent regulatory relationships or are simply driven by global co-occurrence frequencies. To address this, we conducted two complementary analyses: (i) shuffle-based control experiments and (ii) correlation analyses between embedding similarity and simple co-occurrence metrics.

(i) As an illustrative example, we first examined RNF2, which has been reported to participate in both transcriptional activation and repression². ChromBERT embeddings showed markedly higher similarity between RNF2 and its three representative cofactors (PCGF2, MYC and EP300) at distinct subsets of RNF2's binding sites (Figure S3C). To test whether these embedding similarities are learned from global co-occurrence bias, we performed control experiments using shuffled binding data. Specifically, for RNF2's binding sites, we randomly shuffled the binding status of PCGF2, MYC and EP300 across these regions while preserving their overall occurrence frequencies and pairwise co-occurrence rates with RNF2. As shown in Figure R4A, the embedding similarity between RNF2 and

each cofactor decreased significantly after shuffling, indicating that ChromBERT embeddings are sensitive to the the actual combinatorial binding context rather than simply reflecting global co-occurrence frequencies. This analysis is included in the revised manuscript as Figure S3D. In addition, we performed a similar control experiment to 3D chromatin structure. Using CTCF ChIA-PET data in hESCs, we found that the embedding similarity between loop anchors decreased substantially after random shuffling of regulator binding at one anchor when the overall regulator occurrence frequencies remain unchanged (Figure R4B). These control experiments confirm that ChromBERT's representations depend on biologically meaningful regulatory architectures, rather than statistical co-occurrence patterns.

(ii) We further quantitatively compared ChromBERT's embedding similarities to simple co-binding metrics to determine whether embeddings provide additional information beyond statistical co-occurrence. Specifically, for RNF2 and each of ~1,000 other transcription regulators, we computed three pairwise similarity scores: embedding similarity, Jaccard index, and mutual information, based on their binding profiles on chromosome 1. As shown in Figure R4C, we observed moderate correlations between embedding similarity and each of the two co-binding metrics, indicating that the embeddings are partially shaped by direct co-binding patterns. For instance, some RNF2 cofactors (such as CBX6 and RING1) ranked highly in both embedding and co-binding measures. In contrast, more known RNF2 cofactors, especially cofactors involved in its non-classical activation function (e.g., EP300, MYC, MED1, and BRD4), emerged as outliers, showing markedly high embedding similarity with RNF2 despite only moderate co-binding scores. These regulators consistently deviated above the trend lines in the scatter plots, suggesting that ChromBERT captures higher-order, context-dependent regulatory relationships not readily explained by global binding overlap.

Together, these analyses demonstrate that ChromBERT embeddings encode context-specific regulatory architectures and functional cooperativity among transcription regulators, rather than simply reproducing global co-occurrence patterns.

Figure R4. Shuffle experiments and comparison between embedding similarity and simple co-binding metrics.

(A) Boxplots showing cosine similarity between RNF2 embeddings and those of other regulators under original (light green) or shuffled (pink) conditions within subset of RNF2 binding sites. Shuffling was performed simultaneously for PCGF2, EP300, and MYC across the full set of RNF2 representative peaks. This approach preserved their overall occurrence frequencies and pairwise co-occurrence rates with RNF2. (B) Boxplot showing cosine similarity between embeddings of paired loop anchors from K562 CTCF ChIA-PET data (ENCFF118PBQ) under original (light green) or shuffled (pink) conditions. Both loop anchors were required to overlap ATAC-seq peaks (ENCFF558BLC), and only loops spanning 100-200 kb on chromosome 1 were retained ($n = 1,661$). We shuffled the binding status of all cistromes within one loop anchor to assess the effect of co-occurrence patterns on embedding similarity. The statistical significance in (A) and (B) were assessed by a two-sided Student's t -test, and **** represents p -value $< 1 \times 10^{-4}$. (C) Correlation analyses comparing different measures of regulatory cooperation between RNF2 and other transcription regulators on chromosome 1. Each point represents one regulator, with its cooperation with RNF2 quantified by three metrics: embedding similarity, Jaccard index, and Mutual information. Pearson (PCC) and Spearman (SCC) correlation coefficients between embedding similarity and each co-binding metric are indicated. Known RNF2 cofactors are highlighted in dark orange. The dashed trend line denotes $y = x +$ (mean embedding similarity – mean Jaccard index or Mutual information).

4) The manuscript leverages DNase-seq and scGPT-derived transcriptome-based prompts to align cell-type-specific chromatin accessibility and transcriptional states with ChromBERT embeddings, enabling cistrome imputation in unseen cell types. However, it remains unclear why aligning these two

modalities should provide sufficient information to predict cell-type-specific binding sites of transcription factors, especially in cases where chromatin accessibility is similar across cell types.

To strengthen this claim, the authors are encouraged to:

- Provide references to prior studies that demonstrate a strong predictive link between DNase-seq or transcriptomic profiles and transcription factor binding specificity.*
- Perform ablation studies or comparative experiments testing different prompt sources (e.g., DNase-seq only, scGPT only, or combined prompts) to quantify their respective contributions to cistrome imputation.*
- Validate whether the improved performance on cell-type-specific binding sites persists when controlling for chromatin accessibility, by comparing predictions with and without incorporating prompt information.*

Response:

We thank you for these constructive suggestions. In the revised manuscript, we have added references to prior studies demonstrating that both DNase-seq³ (Wang *et al.*, Genome Res, 2012) and transcriptome profiles⁴ (Karimzadeh *et al.*, Genome Biol, 2022) provide predictive information about transcription regulator binding specificity.

In our original study, DNase-seq-derived embeddings and scGPT-derived cell embeddings were used independently as cell-type-specific prompts for ChromBERT to enable cistrome imputation in unseen cell types. This design reflects practical scenarios in which only one data modality (e.g., accessibility or transcriptomics) may be available. We revised the manuscript and corresponding figures to clearly distinguish these two prompt configurations: ChromBERT-D (DNase-seq prompts), and ChromBERT-R (RNA-seq prompts). Following your suggestions, we benchmarked three prompt configurations: (i) DNase-seq prompts (ChromBERT-D), (ii) RNA-seq prompts (ChromBERT-R), and (iii) combined prompts (concatenation of DNase-seq and scGPT embeddings; ChromBERT-combined). As shown in Figure R5, DNase-seq prompts generally achieved better performance than RNA-seq prompts (Figure R5). Specifically, ChromBERT-D outperformed ChromBERT-R in 65 out of 101 test cistromes (AUPRC difference > 0.05), consistent with the well-established relationship between chromatin accessibility and transcription regulator binding. Nonetheless, ChromBERT-R outperformed ChromBERT-D in 8 cistromes (AUPRC difference > 0.05), including EZH2 and REST, which are associated with transcriptional repression and may not directly correlate with open chromatin. This highlights the potential of transcriptome-based prompts to provide complementary information in certain regulatory contexts. Interestingly, the combined prompt configuration did not improve the overall performance beyond that of DNase-seq prompt alone. This may reflect the limitation of simple concatenation strategy, suggesting that more advanced multimodal integration strategies, such as joint representation learning frameworks, may be required to fully leverage the synergistic effects. These benchmarking results are incorporated into the revised manuscript as the updated Figure S4E.

Figure R5. Benchmarking performance for three cell-type-specific prompts for cistrome imputation. Heatmap showing the AUPRC for 101 test cistromes using three prompt configurations: DNase-seq prompts (ChromBERT-D), RNA-seq prompts (ChromBERT-R), and combined prompts (ChromBERT-combined). Cistromes were grouped into three clusters: ChromBERT-D outperformed ChromBERT-R by AUPRC > 0.05, ChromBERT-R outperformed ChromBERT-D by AUPRC > 0.05, and others.

We further evaluated cistrome imputation performance specifically on a subset of cell-type-specific binding sites, which show accessible chromatin in both cell types. As shown in Figure R6, for 78 cell-type pairs across 21 transcription regulators, both DNase-seq and combined prompts achieved significantly better performance than RNA-seq prompts, although all three prompt configurations performed reasonably well in this challenging scenario, in which accessibility patterns alone were uninformative. This result demonstrates that these prompts encode meaningful cell-type-specific information beyond accessibility. While RNA-seq prompts inherently encode cell type information through scGPT-derived cell representations, DNase-seq prompts also capture cell type information via position embeddings. These embeddings allow ChromBERT to distinguish chromatin landscapes between cell types, even when accessibility levels at individual loci appear similar. This explains why DNase-seq prompts can perform well even for binding sites where chromatin accessibility patterns are similar across cell types.

Figure R6. Evaluation of different prompts for imputing cell-type-specific binding at regions with similar chromatin accessibility. Box plots compare the performance of ChromBERT using three prompt configurations: combined prompts (ChromBERT-combined), DNase-seq prompts (ChromBERT-D), and RNA-seq prompts (ChromBERT-R) (left to right). The evaluation included 78 cell-type pairs across 21 transcription regulators. For each pair, the analysis focuses on genomic regions that are accessible in both cell types, but are differentially bound by the transcription regulator. Performance was quantified based on the difference in predictive probabilities between the two cell types to predict the ground-truth cell-type specificity derived from ChIP-seq peaks. Statistical significance was assessed using a two-sided Mann-Whitney *U*-test, with **** indicating p -value $< 1 \times 10^{-4}$ and n.s. indicating non-significant difference.

5) The three downstream tasks (eQTL fine-mapping, enhancer activity prediction, and EZH2 functional mode classification) share a common framework: ChromBERT is first fine-tuned on task-specific objectives to obtain context-adapted TRN embeddings, and then regulator embeddings are interpreted via embedding shift analysis to identify key regulators. While this unified approach is a strength of the study, several aspects require clarification and further validation:

- Clarify methodological consistency and differences. Although the overall pipeline is similar, the manuscript does not explicitly explain whether the same fine-tuning strategy (e.g., loss functions, training epochs, prompt usage) is applied across all tasks or if adjustments are made for each task. A brief comparison of how fine-tuning is tailored to different objectives (classification vs. regression) would improve reproducibility.

Response:

We thank you for highlighting this important point. While the three downstream tasks (eQTL fine-mapping, enhancer activity prediction, and EZH2 functional mode classification) share a unified framework in which ChromBERT is fine-tuned to obtain context-adapted TRN embeddings, we tailored the fine-tuning strategy to each task's specific objectives. The key differences are as follows: (i) loss functions: we applied cross-entropy loss for binary classification tasks (eQTL fine-mapping and EZH2 functional mode classification) and root mean square error (RMSE) loss for the regression task

(enhancer activity prediction); (ii) prompt usage: DNABERT2-derived DNA sequence prompts were incorporated in the eQTL fine-mapping task, whereas the other two tasks did not require prompt inputs; (iii) hyperparameter tuning: while training epochs, learning rates, and batch sizes were tuned within the same search space, they were optimized separately based on validation performance for each task. We added these details in Table S3 and clarified them in the updated Methods section to facilitate reproducibility and improve transparency regarding how ChromBERT was adapted to distinct downstream applications.

- Explain how embedding shift is quantified and validated. The interpretation step is central to all three tasks, yet the manuscript lacks details on how embedding shift is computed (e.g., distance metrics, normalization) and whether results are robust to alternative similarity measures. It would strengthen the claims to include validation experiments, such as comparing embedding shift-derived regulators with known functional regulators from perturbation datasets.

Response:

We thank you for highlighting these important points. For the interpretation step, we quantified embedding similarity and embedding shift using cosine similarity by default, defined as:

$$\text{similarity}(x, y) = \frac{x \cdot y}{\|x\| \|y\|}$$

$$\text{shift}(x, y) = 1 - \text{similarity}(x, y)$$

In this context, embedding shift quantifies the directional divergence between two embedding vectors, thereby reflecting context-dependent regulatory differences under different biological conditions. We applied this framework to each task as follows: (i) eQTL fine-mapping: For each regulator, we calculated the shift of its embedding vectors between causal and non-causal eQTL loci. Regulators were ranked by embedding shift to identify those exhibiting the most context-dependent regulatory roles. (ii) Enhancer activity prediction: We quantified the embedding shifts of cofactors between wild-type and EP300-perturbed conditions at enhancers that showed decreased STARR-seq activity upon EP300 perturbation, enabling detection of cofactors whose function is modulated by EP300. (iii) EZH2 functional mode classification: Instead of shift, we focused on embedding similarity to EZH2 to identify regulators showing strong context-dependent co-association patterns with EZH2. We have clarified the computational procedures in the revised Methods and Figure Legends section.

To assess robustness, we repeated the embedding similarity and shift analyses using three alternative similarity and distance metrics: (i) L1 (Manhattan) distance: each embedding vector was first L1-normalized so that $\sum_i |x'_i| = 1$, and then we calculated shift and similarity as: $\text{shift}(x, y) = \|x' - y'\|_1$, and $\text{similarity}(x, y) = 1 - \text{shift}(x, y)$. (ii) Pearson correlation: we calculated Pearson correlation coefficient r as similarity and $1 - r$ as shift. (iii) Spearman correlation: we calculated Spearman correlation coefficient ρ as similarity and $1 - \rho$ as shift. Across all three metrics, the results were largely consistent with those obtained using cosine similarity (Figure R7), especially for Pearson correlation, indicating that our interpretation is robust to the choice of distance metric. This robustness analysis was added to the updated Figure S8 in the revised manuscript.

A Top 20 differential regulators averaged across 49 tissues.

B Embedding shift of cofactors after EP300 perturbation:

C Embedding similarity between EZH2 and its cofactors:

Figure R7. Interpretation analyses across three downstream tasks using different similarity or distance metrics. Four metrics for quantifying embedding relationships: cosine similarity, L1-normalized L1 distance, Pearson correlation coefficient (PCC), and Spearman correlation coefficient (SCC). (A) Bar plot illustrating the top 20 transcription regulators ranked by embedding shift between causal and non-causal eQTLs, averaged across 49 tissues. Panels (left to right, top to bottom) show four embedding shift measures: 1-cosine similarity, L1 distance, 1-PCC and 1-SCC. (B) Heatmap showing embedding shift for potential cofactors of EP300 at enhancers before and after the perturbation of EP300/CREBBP. Two enhancer groups are shown: activity-decreased enhancers (\log_2 fold change < -1 , $n = 164$) and activity-unchanged enhancers ($-0.5 < \log_2$ fold change < 0.5 , $n = 343$). Panels show four embedding shift measures. (C) Circos plot showing embedding similarities between EZH2 and other regulators at classical (with H3K27me3) and non-classical (without H3K27me3) genomic loci. Two groups of regulators are highlighted: (i) classical group (green) shows higher embedding similarity with EZH2 at classical loci compared to non-classical loci (embedding similarity with EZH2 at classical loci ranking the top 5% among all regulators and the embedding similarity difference between classical loci and non-classical loci > 0.1); (ii) the non-classical group (yellow) exhibits the converse pattern. Each node represents a regulator, and the transparency of edges linking two nodes represent the embedding similarity of two regulators, only edges with high pairwise embedding similarity (> 0.8) were plotted. Panels show four embedding similarity measures.

Following your suggestion, we further conducted external validation using perturbation datasets to assess whether embedding shift could accurately identify the perturbed regulator. We utilized a recently published ATAC-seq dataset containing multiple transcription regulator perturbations⁵, where 5 out of 10 perturbed regulators overlapped with our Cistrome-Human-6K set. For each of these five perturbations, we fine-tuned ChromBERT and ranked regulators by embedding shift between control and perturbed conditions. Encouragingly, in 3 out of 5 perturbations, ChromBERT ranked the correct perturbed regulator within the top 10 candidates based on embedding shift (Figure R8). ChromBERT did not successfully identify EP400 and SMARCA4, both of which are broad chromatin remodelers with global influence on chromatin accessibility. These results demonstrate the biological relevance and partial limitations of our interpretation strategy, and were included as the updated Figure S12G.

Figure R8. Validation of embedding shift-based interpretation using regulator perturbation ATAC-seq datasets. Heatmap showing the top 10 transcription regulators exhibiting the largest embedding shifts between decreased and un-changed or increased regions following perturbation. The perturbed regulators are highlighted in red. Chromatin accessibility changes were analyzed using ATAC-seq datasets in eHAP cell line (GSE144448⁵), including control and knockout samples for CNOT3, EP400, TFDP1, THAP11, and SMARCA4. The top 200 regions showing the greatest decrease or increase in ATAC-seq signal, together with 2,000 un-changed regions, were used for embedding-shift

analysis to identify key regulators, following a strategy similar to that used for driver regulator identification during transdifferentiation.

- Discuss the generalizability of the interpretation framework. While the three tasks demonstrate the versatility of the approach, the authors should discuss potential limitations when applying this pipeline to other tasks or regulators with sparse training data. For example, does the fine-tuning process risk overfitting when only a small set of labeled examples is available?

Response:

We thank you for raising this important point regarding the generalizability and potential limitations of our interpretation framework, especially under data-sparse conditions. To evaluate the generalizability of ChromBERT's fine-tuning framework with limited training data, we conducted down-sampling experiments on two representative downstream tasks: enhancer activity prediction and EZH2 functional mode classification. Specifically, we progressively reduced the size of the training set and evaluated model performance on the validation and test sets. As expected, severe down-sampling increases the risk of overfitting during fine-tuning and leads to decreased performance on validation data (Figures R9A, B), consistent with well-known challenges in deep learning. To reduce the effect of overfitting in the downstream fine-tuning tasks, we chose the checkpoint with the lowest validation loss as the final fine-tuned model. The results show that the framework under this protocol remains largely robust on test data under moderate reductions in training data (Figures R9C, D). We included this analysis in the updated Figure S9 to clarify both the strengths and the boundaries of ChromBERT's generalizability.

Figure R9. Performance of ChromBERT across two downstream tasks using the full training dataset (100%) and down-sampled subsets (75%, 50%, 25%, and 10%). (A) Line plots show the validation RMSE over training epochs for ChromBERT trained on different down-sampled training sets for the task of modeling STARR-seq signals in wild-type HCT116 cells. (B) Line plots show the validation BCE loss over training epochs for ChromBERT trained on different down-sampled training sets for classifying EZH2's classical and non-classical sites. (C) Bar plots show ChromBERT's performance across down-sampled training sets for modeling STARR-seq signal in wild-type HCT116 cells. The notation (n=x) indicates the total number of samples in the training dataset. (D) Bar plots show ChromBERT's performance across down-sampled training sets for classifying EZH2 classical vs non-classical sites. The notation (n=x/y) indicates the number of positive and negative samples in the training dataset.

6) The authors present ChromBERT as a powerful framework for fine-tuning pre-trained TRN embeddings to predict transcriptional regulatory outcomes and identify key regulators through embedding shift analysis.

ChromBERT inherently learns contextual relationships among transcription factors via self-attention mechanisms, with each TRN embedding encoding a region-specific representation of regulator interactions. The regulator embeddings can be regarded as node-specific contextual features, while self-attention weights or embedding similarities may implicitly reflect co-regulatory relationships

between transcription factors. These properties suggest that ChromBERT could be extended beyond key regulator ranking to systematically infer transcriptional regulatory networks (TRNs) at cell-type- or state-specific resolution.

While the results convincingly demonstrate its predictive and interpretive capabilities, the study would benefit from a deeper discussion of ChromBERT's potential for explicit regulatory network reconstruction, which could significantly broaden its impact.

Given that TRN embeddings encode region-specific contextual interactions among transcription factors, several methodological directions could be explored:

Utilizing self-attention weights and embedding similarities - The transformer architecture inherently models dependencies among regulators, and the resulting attention weights or embedding similarity matrices could serve as a basis for inferring regulator-regulator connections. Aggregating these relationships across genomic regions might allow the construction of cell-type-specific regulatory interaction networks.

Combining embeddings with external evidence - To ensure biological relevance and directionality, predicted edges could be filtered or weighted using additional information such as motif presence, ChIP-seq binding profiles, or perturbation-based functional screens.

Response:

We thank you for the positive assessment of ChromBERT's predictive and interpretive performance, and for highlighting the exciting opportunity to extend our framework toward cell-type-specific transcriptional regulatory network (TRN) reconstruction. While the present study primarily focused on predictive accuracy and key regulator identification through embedding similarity and shift analyses, we fully agree that ChromBERT provides a promising foundation for explicit reconstruction of TRNs within specific cellular contexts. Each TRN embedding captures contextual dependencies among transcriptional regulators within a genomic region via the self-attention mechanism, which encodes both pairwise and higher-order regulatory interactions. By aggregating these contextual relationships, derived from embedding similarity matrices or attention-weight distributions, across biologically meaningful genomic regions, ChromBERT could systematically reconstruct accurate and stable hierarchical TRN structures that reflect global co-regulatory patterns. Furthermore, by fine-tuning ChromBERT using chromatin accessibility or transcriptomic profiles from specific cell types, the TRN embeddings become adapted to the corresponding cellular context. This adaptation would refine the learned regulator-regulator interaction structure to reflect cell-type-specific regulatory logic. Such reconstructions would facilitate comparative analyses of regulatory network rewiring across development, differentiation, or disease progression.

We also agree that edges within the regulatory network could be further refined or weighted with external evidence to strengthen its biological interpretability and directionality, for example: (i) motif constraint, retaining only edges where the source regulator has a known binding motif near the target's regulatory sites; (ii) ChIP-seq or ATAC-seq support, retaining only edges supported by experimental binding or accessibility evidence; and (iii) perturbation-based functional screens, validating that perturbing the source regulator alters the activity of the target, which also informs edge directionality. We expanded the Discussion section in the revised manuscript to include these perspectives.

Minor Comments

1) The schematic in Figure 2 is conceptually unclear, particularly in differentiating regulator embeddings from TRN embeddings and showing how these two representations interact during prompt-enhanced fine-tuning. The current layout makes it difficult to follow the flow of information and understand how prompts are integrated into the model to enable regulator-specific cistrome predictions.

To improve clarity, the authors should:

Clearly define and visually distinguish regulator embeddings versus TRN embeddings, explaining their respective roles.

Provide a more structured diagram that explicitly illustrates how prompts are combined with these embeddings during fine-tuning and imputation.

Ensure that the logical flow of the figure aligns with the textual description, making it easier to understand the relationships among prompts, embeddings, and final predictions.

Response:

We thank you for this helpful suggestion. In the revised manuscript, we redesigned Figure 2A (also referred to as Figure R10) to clearly distinguish TRN embeddings and regulator embeddings, and to illustrate how these interact with cell-type-specific prompts during prompt-enhanced fine-tuning. In the cistrome imputation task, each prediction is defined by three attributes: the genomic region, the transcription regulator, and the cell type. The TRN embedding represents the full contextual embedding matrix of all regulators within a given genomic region, capturing region-level regulatory architecture. The regulator embedding, corresponding to a specific transcription regulator extracted from this matrix, represents regulator-specific contextual features at that region. Meanwhile, cell-type-specific prompts, derived from DNase-seq accessibility profiles or scGPT-based cell embeddings, encode cell-type-specific regulatory context. This architecture allows ChromBERT to flexibly combine TRN embeddings, regulator embeddings and prompts, enabling the model to infer regulator binding in unseen cell types or for unseen regulators.

During fine-tuning, TRN embeddings are averaged across the regulator dimension to obtain a region-level vector, and the prediction head receives three concatenated inputs: (i) the averaged TRN embedding vector, (ii) the regulator embedding of interest, and (iii) the prompt embedding. This design allows ChromBERT to integrate both region-specific regulatory context and cell-type-specific conditions to predict whether a particular regulator binds at a specific genomic region in a given cell type. After fine-tuning, cistrome imputation can be performed simply by substituting the corresponding regulator embedding and the prompt for the target cell type, allowing the model to generalize to novel regulator-cell type combinations. The revised schematic now adopts a clear left-to-right logical flow, making the relationships among prompts, embeddings, and final predictions easier to understand.

Figure R10. Schematic illustration of the prompt-enhanced fine-tuning framework for cistrome imputation.

2) The manuscript demonstrates that DNase-seq-based chromatin accessibility and scGPT-derived transcriptome prompts improve ChromBERT's ability to impute cell-type-specific cistromes. However, the relative contributions of chromatin accessibility information and gene expression information to predicting cell-type-specific transcription factor binding remain unclear.

The authors should discuss:

To what extent each modality (DNase-seq vs. transcriptome-based prompts) contributes to model performance, and whether one source provides more cell-type-specific information than the other.

The biological plausibility of these contributions, supported by references to prior studies showing correlations between accessibility, expression, and TF binding.

Potential differences in how these modalities might complement each other, particularly for factors whose binding is not strictly determined by accessibility alone.

Response:

We thank you for highlighting this important issue. Following your suggestions, we benchmarked three prompt configurations: (i) DNase-seq prompts (ChromBERT-D), (ii) RNA-seq prompts (ChromBERT-R), and (iii) combined prompts (concatenation of DNase-seq and scGPT embeddings; ChromBERT-combined). As shown in Figure R5, DNase-seq prompts generally achieved better performance than RNA-seq prompts (Figure R5). Specifically, ChromBERT-D outperformed ChromBERT-R in 65 out of 101 test cistromes (AUPRC difference > 0.05), consistent with the well-established relationship between chromatin accessibility and transcription regulator binding. Nonetheless, ChromBERT-R outperformed ChromBERT-D in 8 cistromes (AUPRC difference > 0.05), including EZH2 and REST, which are associated with transcriptional repression and may not directly correlate with open chromatin. This highlights the potential of transcriptome-based prompts to provide complementary information in certain regulatory contexts. Interestingly, the combined prompt configuration did not improve the overall performance beyond that of DNase-seq prompt alone. This may reflect the limitation of simple concatenation strategy, suggesting that more advanced multimodal

integration strategies, such as joint representation learning frameworks, may be required to fully leverage the synergistic effects. For the biological plausibility of these insights, we have added references in the revised manuscript to prior studies demonstrating that both DNase-seq³ and transcriptome profiles⁴ provide predictive information about transcription regulator binding specificity.

3) Figure 3A provides an overview of how fine-tuned TRN embeddings are used for interpretation via embedding shift, but the schematic is highly abstract and does not clearly illustrate the workflow for each downstream task (eQTL fine-mapping, enhancer activity prediction, EZH2 functional mode classification).

To improve clarity, the authors could extend Figure 3A or provide supplementary subpanels that explicitly depict the task-specific inputs and labels used for fine-tuning in each task (e.g., causal/non-causal variants for eQTL, enhancer activity values for STARR-seq).

Such a figure would better illustrate the versatility of the approach and help readers understand the specific workflows for each application.

Response:

We thank you for this constructive suggestion. In the revised manuscript, we added a new supplementary figure to explicitly illustrate the task-specific fine-tuning workflows (Figure S6; also referred to as Figure R11). The figure now outlines the fine-tuning model architecture, fine-tuning objectives, and labels used for the three downstream applications: (i) eQTL fine-mapping, a binary classification task distinguishing causal versus non-causal variants, with fine-tuning based on tissue-specific TRN embeddings and DNABERT2-derived sequence prompts; (ii) enhancer activity prediction, a regression task where TRN embeddings are used to model quantitative enhancer activity values from STARR-seq assays; (iii) EZH2 functional mode classification, a binary classification task distinguishing classical versus non-classical EZH2 binding sites. While the overall architecture remains consistent across tasks, the figure highlight task-specific variations, such as the inclusion of sequence prompts for eQTL fine-mapping. Together, this figure makes the workflow of each downstream task more explicit and highlight the versatility of ChromBERT as a unified framework for diverse regulatory prediction objectives.

Figure R11. Schematic illustration of the workflow for the three fine-tuning downstream tasks.

Reviewer #2:

The work describes ChromBERT, a deep learning model of chromatin state trained in a self-supervised fashion to predict masked tokens within 1 kb regions in the human genome. The resulting model is evaluated on its ability to serve as a platform for training accurate models for other regulatory genomics applications. The approach is novel and is complementary to the more traditional sequence-based approach.

Strengths:

* Well thought out deep learning approach that is unique in its modeling approach. The idea of adding additional external data they refer to as "prompts" is also novel, and allows the model to incorporate complementary information e.g. from DNA LLMs.

* Comprehensive evaluation on several gene regulation tasks demonstrate the advantage of fine-tuning ChromBERT over fine-tuning sequence-based models (Enformer and DNABERT2).

* The manuscript is well-written and relatively easy to understand (except when it isn't - see comments below).

Weaknesses:

* Some aspects of the model need to be explained better. For example, the description of the decoder architecture during pre-training; Hi C imputation also needs to be explained better. In this case, it's not clear how the CNN and genomic distance map were used (and why a power of -0.75, which seems rather arbitrary).

Response:

We thank you for pointing out the need for clearer explanations. In the revised Methods section, we expanded our description of the decoder architecture used during pre-training to clarify its structure and function in reconstructing masked tokens. Specifically, ChromBERT employs a decoder architecture that follows the encoder to reconstruct input binding status vector X_i from the TRN embedding $E_i^{(TRN)} \in \mathbb{R}^{P \times d_{model}}$, producing an output $\hat{Y}_i \in \mathbb{R}^{P \times 5}$. The decoder is a two-layer multilayer perceptron (MLP): the first MLP layer applies a linear transformation with parameters of shape (d_{model}, d_{model}) , followed by a ReLU activation; the second MLP layer then projects the hidden dimension to the output vocabulary dimension with parameters of shape $(d_{model}, 5)$.

We also provide more details for the Hi-C imputation task in the revised Methods section. In this task, we used pre-trained TRN embeddings to impute high-resolution contact maps from low-resolution Hi-C contact map with a CNN. The input for each region pair consists of three channels:

- an O/E-normalized low-resolution Hi-C contact map (up-sampled by bilinear interpolation from 5-kb resolution to 1-kb resolution);
- a pairwise TRN embedding similarity map computed by cosine similarity of pre-trained TRN embeddings for regions pairs;

- a pairwise genomic distance map expressed as $(1 + D_g)^{-0.75}$, where D_g represents genomic distance of a region pair, and the exponent -0.75 is grounded in prior work modeling chromatin contact decay with genomic distance^{6,7}.

These three channels are stacked into an input tensor of shape $(3, N, N)$, where $N = 1,000$ (a 1-Mb genomic window at 1-kb resolution). The CNN has three 2D convolution layers: the first expands the three input channels into 64 feature channels using 64 kernels of size 13×13 (with padding to preserve spatial dimensions); the second further refines these 64 channels using the same kernel size; and the final layer applies a single 13×13 kernel to collapse the representation into one channel, producing the predicted contact map of shape $(1, N, N)$.

During training, we used a sliding window of 1-Mb (1,000 1-kb bins) moving diagonally in 200-kb steps. To match the target resolution of 1 kb, Hi-C contact maps originally processed at a 5-kb resolution were up-sampled to 1-kb by using two-dimensional linear interpolation. Specifically, we applied bilinear interpolation, which estimates contact values at intermediate 1-kb positions by linearly combining the values of the four nearest bins in both row and column directions⁸. Ground truth was provided by O/E-normalized Micro-C contact maps. Chromosomes were split into training (1, 4, 7, 10, 13, 17, 18), validation (3, 6, 9, 12, 16, 19, 20, X), and test (2, 5, 8, 11, 14, 15, 21, 22) sets. Finally, to reduce bias from regions excluded during ChromBERT pre-training (where regulator binding was assumed all-zero), we adjusted cosine similarity values by replacing them with the average similarity at the same genomic distance.

These clarifications are now incorporated into the revised Methods section. We believe they enhance the transparency and clarify the rationale behind key design choices in both the pre-training and downstream Hi-C imputation stages of ChromBERT.

** For the EZH2 task, "ChromBERT was fine-tuned by omitting cistromes associated with H3K27me3 from input reference cistromes to diminish its dominant role in the task." That may not be sufficient, as the model will have been trained on those cistromes, and so is given an unfair advantage over DNABERT2 and Enformer (see also the comment on training, validation and test sets).*

Response:

We thank you for raising this important point. For the EZH2 functional mode classification task, our rationale for omitting cistromes associated with H3K27me3 during fine-tuning was to reduce the direct influence of this dominant histone mark, encouraging the model to rely on contextual regulatory information rather than a single strong signal. However, we agree that ChromBERT was pre-trained on large-scale cistrome data that included H3K27me3, and therefore may retain latent information from H3K27me3, potentially conferring an advantage over baseline models like DNABERT2 and Enformer. To directly address this concern, we additionally pre-trained a version of ChromBERT using only transcription regulator binding data, completely excluding all epigenetic modifications from the pre-training inputs. This version of ChromBERT was then fine-tuned and evaluated on the EZH2 functional mode classification task under the same protocol. Despite the removal of all histone modification data during pre-training, this TF-only ChromBERT still outperformed DNABERT2 and Enformer on this classification task (Figure R12), confirming that its advantage derives from the

contextual embeddings learned during pre-training rather than residual information from H3K27me3. This analysis has been included in the updated Figure S7G.

Figure R12. Performance comparisons of different models on EZH2's dual function classification. Bar plots show the performance of ChromBERT, ChromBERT (pre-trained without epigenetic modifications), ChromBERT (without pre-training), Enformer, and DNABERT-2 in classifying EZH2 binding sites as classical (H3K27me3-associated) or non-classical (H3K27me3-independent). ChromBERT was fine-tuned by omitting cistromes associated with H3K27me3 from input reference cistromes to diminish its dominant role in the task. ChromBERT (pre-trained without epigenetic modifications) is a pre-trained version of ChromBERT pre-trained only on transcription regulator cistromes, excluding all epigenetic modification tracks including H3K27me3, to eliminate any potential advantage from prior exposure to these signals.

** Functional collaboration between regulators is assessed by the similarity of their embeddings. While this heuristic is intuitive, there is nothing in the network that would lead this to hold in general. A common way to establish associations between regulators uses the attention matrix. This has been demonstrated to work well in the context of sequence-based deep learning, and is likely to work here as well. See e.g.:*

*Ullah, Fahad, and Asa Ben-Hur. "A self-attention model for inferring cooperativity between regulatory features." *Nucleic acids research* 49.13 (2021): e77-e77.*

*Zhang, Zhenhao, et al. "A generalizable framework to comprehensively predict epigenome, chromatin organization, and transcriptome." *Nucleic Acids Research* 51.12 (2023): 5931-5947.*

Response:

We thank you for this insightful suggestion and for pointing out relevant prior work. In this study, we used embedding-based interpretation rather than attention matrices-based interpretation. Attention matrices can, in principle, reflect how information flows between tokens and capture pairwise dependencies within a head or a single transformer layer. Several studies have used attention patterns to infer cooperative relationships between regulatory features in sequence-based models⁹⁻¹¹. However, attention matrices describe layer- and head-specific weighting during intermediate message passing and are thus highly sensitive to model depth, initialization, and training noise. Their interpretation

depends on which layer or head is analyzed, making them difficult to compare across models or biological contexts. In contrast, the final regulator embeddings in ChromBERT are derived from iterative multi-head self-attention, residual aggregation, and nonlinear transformations across all layers. These embeddings integrate diverse contextual cues into a single stable latent representation. As such, embedding similarity provides a more robust and biologically interpretable measure of regulator relationships. We have clarified this rationale and discussed the previous studies in the revised Methods sections.

To directly compare these strategies, we also explored attention-based interpretability by max-pooling attention weights across all heads within the final transformer block. While this attention-based method successfully recovered globally cooperative cofactors, it failed to delineate specific regulatory patterns. For example, in the EZH2 dual-function task, attention-based analysis correctly identified cofactors associated with both classical and non-classical EZH2 functions but was unable to distinguish between these two biological modes, resulting in mixed clusters of cofactors (Figure R13A). In addition, we benchmarked attention-based and embedding-based interpretability in identifying key regulators involved in transdifferentiation. Embedding-based interpretation correctly recovered 8 out of 11 known drivers, outperforming attention-based analysis (5 out of 11) under the same fine-tuned models (Figure R13B). These results suggest that embedding-based interpretation provides superior task resolution and biological specificity across diverse regulatory contexts.

Figure R13. Benchmarking attention-based versus embedding-based interpretability. Attention weights were extracted from the final model layer and combined across heads using a max-pooling strategy. (A) Circos plots showing attention weights (left) and embedding similarities (right) with EZH2 for other regulators at classical (with H3K27me3) and non-classical (without H3K27me3) genomic loci. The classical group was marked in green and the non-classical group was marked in yellow. The known cofactors in classical function of EZH2 were highlighted in blue, and the known cofactors in non-classical function of EZH2, E2F1 and STAT3, were highlighted in red. Each node represents a regulator, and the transparency of edges linking two nodes represents attention weights or the embedding similarity of two regulators. (B) Heatmap showing the ranks of known driver regulators across five transdifferentiation transitions in ChromBERT, based on two interpretability approaches: attention-based and embedding-based analyses. Each row represents a known regulator involved in a specific transdifferentiation process, with numeric ranks annotated for those appearing within the top 25. In the attention-based approach, we computed each regulator's total attention weight by using the given regulator as the key and all other regulators as the queries and then summed for different pairs. We

then averaged each regulator's total attention weight between two sets of regions, those showing decreased ATAC-seq signals and those with unchanged accessibility (the same sets used in the embedding-based analysis). Regulators were finally ranked according to the shift between these two averaged attention weights, following the same procedure as in the embedding-based approach. The five transitions and their known driver regulators are: fibroblasts to iPSCs (POU5F1, SOX2, NANOG); B cells to macrophages (CEBPA, SP1); fibroblasts to myoblasts (MYOD1); fibroblasts to hepatocytes (HNF1A, HNF4A); and fibroblasts to cardiac myocytes (GATA5, TBX5, MEF2C).

** Sequence based models generate embeddings that are useful for tasks across species. It is not clear if the embeddings generated by ChromBERT (trained in human) are useful in other species, e.g. mouse.*

Response:

ChromBERT was pre-trained on large-scale human cistrome data, and its embeddings are therefore optimized for human regulatory contexts. We acknowledge that the current version does not directly enable cross-species transfer. To address this limitation, we have also pre-trained a mouse-specific version of ChromBERT using publicly available mouse cistrome datasets. This mouse version ChromBERT is available through our GitHub repository and supports species-specific applications. This extension demonstrates the adaptability of the ChromBERT framework to other species when trained on appropriate data.

We also appreciate your suggestion to consider broader cross-species generalization. While the current study focuses on modeling transcription regulator binding profiles within a single species, future versions could achieve cross-species transfer by integrating DNA sequence-based foundation models to capture conserved regulatory grammars. Evolutionarily conserved sequences provide a biologically meaningful basis for aligning regulatory representations between organisms. Coupling ChromBERT's regulator-centered embeddings with sequence-level representations from DNA foundation models may therefore enable cross-species inference of transcriptional regulatory networks, particularly between closely related species such as human and mouse. We have highlighted this perspective in the revised Discussion section.

Additional Comments:

** It is interesting that ChromBERT outperforms Enformer, which is trained on very long sequences, whereas ChromBERT is trained on short 1 kb regions. Please comment on that!*

Response:

We thank you for this insightful comment. ChromBERT outperforms Enformer despite being trained on shorter 1-kb genomic regions because it captures a distinct layer of regulatory information, i.e., context-dependent interactions among transcription regulators, that complements sequence-based features. While models such as Enformer primarily learn sequence grammar and long-range dependencies directly from DNA, ChromBERT encodes regulator-regulator relationships derived from large-scale cistrome data. These embeddings capture the combinatorial and context-specific cooperation among transcription regulators that underlies transcriptional regulation, enabling ChromBERT to infer functional architectures even without modeling long DNA sequences. In other words, ChromBERT shifts the focus from sequence-driven modeling to regulatory-context modeling,

leveraging the latent structure of transcriptional regulatory networks (TRNs) to generalize across cell types and regulatory conditions.

Importantly, ChromBERT and sequence-based models such as Enformer or DNABERT-2 are conceptually complementary. Sequence-based models learn how regulatory potential emerges from DNA motifs and chromatin context, whereas ChromBERT models how transcription regulator interpret and execute these regulatory instructions. Integrating both sequence-level and regulator-level modalities represents a promising direction for future multimodal foundation models in regulatory genomics. We emphasized this point in the revised Discussion section.

** A comparison with the "Sei" model for example, might be instructive (but not necessary) since it is trained on much shorter sequences, and its training data also consists of a similar mix of chromatin state profiles.*

Response:

We thank you for this helpful suggestion. In response, we included the Sei model in our benchmarking analysis. While Sei is trained on shorter genomic sequences and integrates diverse chromatin state profiles, it did not outperform ChromBERT, nor did it surpass Enformer in our evaluation (Figure R14). These findings further highlight ChromBERT's advantage in capturing context-specific regulatory interactions beyond chromatin state features alone.

Figure R14. Performance comparison of ChromBERT and baseline models across downstream fine-tuning tasks. (A) Box plots comparing the performance in classifying causal versus non-causal eQTLs across 49 human tissues for ChromBERT, ChromBERT (without pre-training), DNABERT2, Sei, and Enformer, as indicated by AUROC. (B) Bar plots illustrate the performance of ChromBERT, ChromBERT (without pretraining), Enformer, Sei, and DNABERT-2 in modeling STARR-seq signals in wild type HCT116 cells. (C) Bar plots show the performance of ChromBERT, ChromBERT (without pre-training), Enformer, Sei, and DNABERT-2 in classifying EZH2's classical and non-classical sites.

** Please add running times of fine-tuning ChromBERT in comparison with time for fine-tuning other models.*

Response:

We thank you for this suggestion. In the revised manuscript, we added a detailed comparison of fine-tuning times between ChromBERT and DNABERT-2 across multiple downstream tasks (Table S4). Fine-tuning time is not reported for Enformer, because its embeddings are fixed and the downstream task is performed using a random forest classifier, making its computational cost primarily determined by the initial embedding extraction step.

As expected, ChromBERT requires longer fine-tuning time than DNABERT-2, owing to its more complex input representation, which integrates over 6,000 cistromes per genomic region. This design enables ChromBERT to explicitly model context-dependent transcriptional regulatory networks (TRNs) by capturing cooperative and competitive interactions among transcription regulators. Although reducing the number of input cistromes could markedly decrease both pre-training and fine-tuning costs, such simplification would also weaken the model's capacity to capture comprehensive regulator-level dependencies and diminish its biological interpretability. We therefore view this trade-off between computational efficiency and representational richness as justified by the biological insight it provides.

** The decay in embedding similarity with distance shown in figure S2B seems quite minor. Please provide details in the manuscript on how that was quantified. Is it the average cosine similarity over all pairs of embedding vectors with a distance falling within a given bin? If that is the case, it would seem all the embeddings are quite similar. Vectors in high dimensional spaces tend to exhibit very low similarity in general, so the level of overall similarity shown in the figure is very surprising.*

Response:

We thank you for these thoughtful comments. In ChromBERT, each genomic region is represented by a transcriptional regulatory network (TRN) embedding matrix of size (number of regulators × hidden dimension). Each row in this matrix corresponds to a regulator embedding vector that encodes the context-dependent role of that regulator after integrating information from all others through the self-attention mechanism. To facilitate region-level analysis, we computed an averaged TRN embedding vector (length = hidden dimension) by averaging TRN embeddings across the regulator dimension for each genomic region. For the decay analysis shown in original Figure S2B, we calculated the cosine similarity between pairs of these region-level averaged TRN embedding vectors as a function of genomic distance. Specifically, all region pairs separated by a given genomic distance were grouped into bins, and we reported the average cosine similarity across all pairs within each bin. This procedure quantifies how embedding similarity decreases as genomic distance increases. We have updated the Methods section and Figure Legends to clarify this quantification procedure.

Regarding the overall high level of similarity, this outcome is expected for two reasons. First, mean pooling across regulators inherently reduces embedding variance, resulting in smoother and more homogeneous region-level representations. The subsequent averaging of cosine similarities across distance bins further attenuates apparent variability in the decay analysis. Second, ChromBERT embeddings capture regulatory architectural similarity rather than random high-dimensional variance. Because a large proportion of the genome consists of inactive regions with limited transcription regulator binding, many genomic regions share broadly similar regulatory contexts with low

We thank you for raising this point. Unlike DNA sequence models where positional embeddings encode sequential order, in our framework, positional embeddings are not used to represent spatial or sequential relationships. Instead, they serve primarily as learnable identifiers to distinguish individual cistromes within the input embedding matrix. We have clarified this rationale in the revised Methods section.

** The github repository contains a mouse version of the model. I did not see it mentioned in the manuscript. Please provides some details on that model!*

Response:

We thank you for raising this point. We also trained a mouse version of ChromBERT using a curated mouse cistrome dataset. This model follows the same pre-training architecture and objectives as the human version but uses publicly available mouse cistrome datasets. It is included in our GitHub repository to support species-specific applications, We have revised the Data and Code Availability section to introduce this mouse model.

Typos and grammar:

** PyTorch Lightning (v2.0.4) was implemented... I think you meant "used"*

** "In our study, we also compared the performance of ChromBERT with two advanced deep language models, Enformer and DNABERT-2" - Language models are trained in a self-supervised fashion to predict masked tokens. The Enformer is trained to predict gene expression, so is not a language model.*

Response:

We thank you for pointing out these typos and terminology issues. In the revised manuscript, we have corrected those phrasing.

References

1. Diao, Y., Fang, R., Li, B., Meng, Z., Yu, J., Qiu, Y., Lin, K.C., Huang, H., Liu, T., Marina, R.J., et al. (2017). A tiling-deletion-based genetic screen for cis-regulatory element identification in mammalian cells. *Nat Methods* 14, 629-635. 10.1038/nmeth.4264.
2. Hu, S., Huo, D., Yu, Z., Chen, Y., Liu, J., Liu, L., Wu, X., and Zhang, Y. (2020). ncHMR detector: a computational framework to systematically reveal non-classical functions of histone modification regulators. *Genome Biol* 21, 48. 10.1186/s13059-020-01953-0.
3. Wang, J., Zhuang, J., Iyer, S., Lin, X., Whitfield, T.W., Greven, M.C., Pierce, B.G., Dong, X., Kundaje, A., Cheng, Y., et al. (2012). Sequence features and chromatin structure around the genomic regions bound by 119 human transcription factors. *Genome Res* 22, 1798-1812. 10.1101/gr.139105.112.
4. Karimzadeh, M., and Hoffman, M.M. (2022). Virtual ChIP-seq: predicting transcription factor binding by learning from the transcriptome. *Genome Biol* 23, 126. 10.1186/s13059-022-02690-2.

discriminative power, leading to higher cosine similarity values than would be expected from random vectors in the latent space. This is a biological rather than an artifact-driven outcome, and we incorporated this explanation in the revised manuscript.

** There are several sequence-based deep learning models that are designed to generalize to unseen cell types, and are worth mentioning:*

*Javed, Nauman, et al. "A multi-modal transformer for cell type-agnostic regulatory predictions." *Cell Genomics* 5.2 (2025).*

The name of the model (EpiBERT) is VERY similar to ChromBERT, and reflects some similarity in the approach.

*Zhang, Zhenhao, et al. "A generalizable framework to comprehensively predict epigenome, chromatin organization, and transcriptome." *Nucleic Acids Research* 51.12 (2023): 5931-5947.*

Response:

We thank you for highlighting these relevant studies. We have carefully reviewed EpiBERT¹² (Javed *et al.*, *Cell Genomics*, 2025) and EPCOT¹¹ (Zhang *et al.*, *Nucleic Acids Res.*, 2023) and introduced both models in the revised Introduction section. Both EpiBERT and EPCOT are sequence-based, multimodal transformers that integrate DNA sequence and chromatin accessibility to generalize across cell types, focusing on learning regulatory sequence grammar and chromatin-state features. In contrast, ChromBERT is regulator-centric, pre-trained on large-scale cistrome data to model context-dependent interactions among transcription regulators, thereby representing transcriptional regulatory networks directly rather than inferring them from sequence features. This design provides complementary strengths: while sequence-based models excel at predicting chromatin or transcriptional outputs from primary sequence, ChromBERT emphasizes interpretable TRN-level representations that capture regulator cooperation and context-specific regulation. We view these approaches as synergistic rather than overlapping, and we highlight in the revised Discussion that integrating sequence-based and regulator-based representations represents a promising future direction for multimodal regulatory genomics.

** There is no clear description of the choice of training, validation, and test sets in most cases. It is important that all tasks use consistent choices in this regard to avoid information leakage.*

Response:

We thank you for this important comment. For all fine-tuning tasks, we randomly split the samples into non-overlapping training, validation, and test sets to prevent information leakage. All models in benchmark experiments were trained and evaluated using the same data partitions to ensure fair comparisons. The details of data splitting for each fine-tuning task were summarized in Table S3.

** Please clarify the reasoning for using positional embeddings on top of categorical embeddings. For DNA sequences it helps establish an order relationship; in this case the notion of order is not present, so the rationale is not clear. Is this simply to establish some identifier for each element?*

Response:

Referees' report, second round of review

Reviewer 1:

The authors have well addressed all my concerns. I have no further comments.

Reviewer 2:

The authors have addressed all my concerns with an in depth response. A few minor comments remain, mostly related to clarity of presentation. Most importantly, I suggest to explain in broad terms how ChromBERT works in the Results section, since it is quite different than other genomics foundation models.

Minor comments:

* I would suggest to the authors to consider a catchier title, along the lines of the title on the github repository: "ChromBERT: A foundation model for learning interpretable representations for context-specific transcriptional regulatory networks"

* For someone reading the paper starting with the results section it is hard to piece together the idea behind ChromBERT, as it works differently than the standard approach of sequence-based models. So I **strongly** suggest to provide more background on how the model works in the Results section. I had to read the Methods section before I could get a good idea of what's going on.

* The HuggingFace webpage for chrombert appears non-functional.

* In the summary the authors state that the output of their model can be used "to reveal mechanistic roles of transcription regulators". Biologists may object to the use of the term "mechanistic", as it suggests a more detailed description than that provided by your model. I would not consider a TRN a full mechanistic model, just a first step towards that.

* "enabling precise interpretation of genome wide regulatory roles" - The word precise oversells what your model is able to do in my opinion.

* It's not clear what can be gleaned from Figure S3C.

* From figure S3B, embedding similarity is only mildly related to gene ontology co-membership, and more strongly to PPIs and membership in hallmark gene sets. Please comment on that.

* I had trouble understanding how the baseline method for cistrome imputation works.

typos and grammar:

* "Transcriptional regulatory network" is more grammatically correct than "Transcription regulatory network".

* regulator cooperations  regulatory cooperation

* representations then can  representations can then

* which together forming  which together form

* "with its performance changed little"

Authors' response to the second round of review

Reviewers' Comments:

Reviewer #1: The authors have well addressed all my concerns. I have no further comments.

We appreciate your positive evaluation and thank you for the constructive comments provided during the revision process.

Reviewer #2: The authors have addressed all my concerns with an in depth response. A few minor comments remain, mostly related to clarity of presentation. Most importantly, I suggest to explain in broad terms how ChromBERT works in the Results section, since it is quite different than other genomics foundation models.

Minor comments:

** I would suggest to the authors to consider a catchier title, along the lines of the title on the github repository: "ChromBERT: A foundation model for learning interpretable representations for context-specific transcriptional regulatory networks"*

We thank you for this excellent suggestion. We agree that the title used in our GitHub repository is more compelling and accurate. We have updated the manuscript title to: "ChromBERT: A foundation model for learning interpretable representations for context-specific transcriptional regulatory networks".

** For someone reading the paper starting with the results section it is hard to piece together the idea behind ChromBERT, as it works differently than the standard approach of sequence-based models. So I **strongly** suggest to provide more background on how the model works in the Results section. I had to read the Methods section before I could get a good idea of what's going on.*

We appreciate this valuable suggestion. We agree that it is crucial to clarify how ChromBERT differs from conventional sequence-based models. Therefore, we have included a brief background description at the start of the Results section (specifically in the new 'Design' subsection) to explain the model's architecture. This addition ensures readers have sufficient conceptual grounding without strictly requiring them to read the Methods first.

** The HuggingFace webpage for chrombert appears non-functional.*

We thank you for bringing this to our attention. We have resolved the issue, and the HuggingFace webpage for ChromBERT is now fully operational.

** In the summary the authors state that the output of their model can be used "to reveal mechanistic roles of transcription regulators". Biologists may object to the use of the term "mechanistic", as it suggests a more detailed description than that provided by your model.*

I would not consider a TRN a full mechanistic model, just a first step towards that.

We appreciate this constructive comment. We have revised the statement in the Summary to “infer regulatory roles of transcription regulators”, which we believe provides a more accurate and appropriately description of the model’s capabilities.

** “enabling precise interpretation of genome wide regulatory roles” - The word precise oversells what your model is able to do in my opinion.*

We thank you for raising this point. We have removed the word “precise” to avoid overselling in the revised manuscript.

** It’s not clear what can be gleaned from Figure S3C.*

We thank you for this helpful comment. We have revised the text accompanying Figure S3C to clarify the purpose of the analysis and the biological insight it provides. Specifically, we now explain that this figure examines genomic-context specificity in regulator embeddings. As shown in the revised description, RNF2, a core PRC1 component, exhibits high embedding similarity to classical repressors (e.g., PCGF2) in repressive chromatin regions, but shifts toward activators (e.g., EP300 and MYC) in regions associated with transcriptional activation. This context-dependent switching mirrors RNF2’s dual functions and demonstrates that the learned regulator embeddings capture biologically meaningful, context-specific regulatory associations¹. We believe this clarification makes the interpretation of Figure S3C much more explicit.

** From figure S3B, embedding similarity is only mildly related to gene ontology co-membership, and more strongly to PPIs and membership in hallmark gene sets. Please comment on that.*

Embedding similarity in our model arises from regulators’ co-binding patterns and therefore reflects their functional coordination in chromatin. Because cooperative binding is often mediated by direct or proximal physical interactions, it is expected that embedding similarity shows its strongest relationship with known protein-protein interactions. Regarding functional annotations, hallmark gene sets represent compact, manually curated biological programs that are highly enriched for transcriptional regulation². In contrast, Gene Ontology categories are considerably broader, more heterogeneous, and not specifically organized around regulatory co-binding or chromatin-level cooperation. As a result, GO co-membership provides a weaker and noisier reference for the type of regulatory interactions captured by our embeddings. Thus, the stronger correspondence with PPIs and hallmark gene sets, and the more modest association with GO terms, reflects the biological specificity of the embedding space rather than a limitation of the method.

** I had trouble understanding how the baseline method for cistrome imputation works.*

We thank you for pointing out the lack of clarity in our description of the baseline method. In response, we have substantially revised and clarified the corresponding section in the Methods. Specifically, we (i) replaced the term "putative peaks" with the more precise term "candidate peaks," (ii) explicitly detailed how candidate peaks are defined under two scenarios, either by merging peaks from at least five existing cistromes of the same regulator, or, when fewer than five are available, by identifying 1-kb bins containing the regulator's motif, and (iii) clearly described how these candidate peaks are weighted by cell-type, specific DNase-seq signal to derive binding probability ranks in the baseline method. These revisions improve the conceptual clarity and readability of the baseline description.

typos and grammar:

** "Transcriptional regulatory network" is more grammatically correct than "Transcription regulatory network".*

** regulator cooperations  regulatory cooperation*

** representations then can  representations can then*

** which together forming  which together form*

** "with its performance changed little"*

We thank you for pointing out these typos and grammar issues. In the revised manuscript, we have corrected those phrasing.

References

1. Hu, S., Huo, D., Yu, Z., Chen, Y., Liu, J., Liu, L., Wu, X., and Zhang, Y. (2020). nCHMR detector: a computational framework to systematically reveal non-classical functions of histone modification regulators. *Genome Biology* 21, 48. 10.1186/s13059-020-01953-0.
2. Liberzon, A., Birger, C., Thorvaldsdottir, H., Ghandi, M., Mesirov, J.P., and Tamayo, P. (2015). The Molecular Signatures Database (MSigDB) hallmark gene set collection. *Cell Syst* 1, 417-425. 10.1016/j.cels.2015.12.004.